# Replisome loading reduces chromatin motion independent of DNA synthesis

Maruthi Kumar Pabba[1†], Christian Ritter[2†], Vadim O Chagin[1,3†], Janis Meyer[2], Kerem Celikay[2], Jeffrey H Stear[4], Dinah Loerke[5], Ksenia Kolobynina[1], Paulina Prorok[1], Alice Kristin Schmid[2], Heinrich Leonhardt[6], Karl Rohr[2*], M Cristina Cardoso[1*]

[1]Department of Biology, Technical University of Darmstadt, Darmstadt, Germany; [2]Biomedical Computer Vision Group, BioQuant, IPMB, Heidelberg University, Heidelberg, Germany; [3]Institute of Cytology RAS, St. Petersburg, Russian Federation; [4]EMBL Australia Node in Single Molecule Science, University of New South Wales, Sydney, Australia; [5]Department of Physics & Astronomy, University of Denver, Denver, United States; [6]Department of Biology II, Ludwig Maximilians University, Munich, Germany

*For correspondence:
K.Rohr@dkfz-heidelberg.de (KR);
cardoso@bio.tu-darmstadt.de
(MCC)

†These authors contributed
equally to this work

Competing interest: The authors
declare that no competing
interests exist.

Reviewing Editor: Volker
Dötsch, Goethe University,
Germany

**Abstract** Chromatin has been shown to undergo diffusional motion, which is affected during gene transcription by RNA polymerase activity. However, the relationship between chromatin mobility and other genomic processes remains unclear. Hence, we set out to label the DNA directly in a sequence unbiased manner and followed labeled chromatin dynamics in interphase human cells expressing GFP-tagged proliferating cell nuclear antigen (PCNA), a cell cycle marker and core component of the DNA replication machinery. We detected decreased chromatin mobility during the S-phase compared to G1 and G2 phases in tumor as well as normal diploid cells using automated particle tracking. To gain insight into the dynamical organization of the genome during DNA replication, we determined labeled chromatin domain sizes and analyzed their motion in replicating cells. By correlating chromatin mobility proximal to the active sites of DNA synthesis, we showed that chromatin motion was locally constrained at the sites of DNA replication. Furthermore, inhibiting DNA synthesis led to increased loading of DNA polymerases. This was accompanied by accumulation of the single-stranded DNA binding protein on the chromatin and activation of DNA helicases further restricting local chromatin motion. We, therefore, propose that it is the loading of replisomes but not their catalytic activity that reduces the dynamics of replicating chromatin segments in the S-phase as well as their accessibility and probability of interactions with other genomic regions.

## eLife assessment

This is a **valuable** investigation of the chromatin dynamics throughout the cell cycle by using fluorescence signals and patterns of GFP-PCNA and CY3-dUTP, which labels newly synthesized DNA. The authors report reduced chromatin mobility in S relative to G1 phase. The technology and methods used are **solid**. The data will be of interest to researchers working on chromatin dynamics.

## Introduction

Dynamic yet functionally stable organization of cellular processes is a crucial feature of biological systems, which allows them to respond to external stimuli and survive. The eukaryotic nucleus is a complex subcellular organelle where DNA metabolism, including its replication, repair, and

transcription, occurs. Eukaryotic DNA is organized in the nuclear space by interactions with histones and architectural proteins to form a hierarchy of domains and compartments of the interphase chromatin. Nuclear architecture is dynamically modulated due to the binding of biomolecules and epigenetic changes of the chromatin. It is also interdependent with DNA metabolism mediated by the action of enzymes on the chromatin. The maintenance of the DNA (including its replication and repair) and its transcription into RNA are spatio-temporally organized within the cell nucleus.

Analysis of the local chromatin dynamics in live cells revealed that an essential aspect of interphase chromatin is its mobile nature (*Gasser, 2002*; *Marshall et al., 1997*). The movement of chromatin loci was shown to be consistent with an anomalous (constrained) diffusion model (*Scipioni et al., 2018*; *Shukron et al., 2019*). This model indicates that a single chromatin locus is corralled within a sub-micron radius and exhibits random diffusion motion and will execute multiple random jumps into neighboring compartments (*Bronshtein et al., 2016*; *Chubb et al., 2002*; *Heun et al., 2001*; *Levi et al., 2005*; *Marshall et al., 1997*). This behavior, which we refer to as local chromatin diffusion (LCD), has been described in multiple systems, suggesting that it is likely to represent a fundamental aspect of chromatin dynamics in eukaryotes.

According to the current paradigm, the 4D organization of the chromatin inherently includes its physical properties as a long polymer (*Esposito et al., 2021*; *Esposito et al., 2019*), while stochastic thermodynamically driven events are likely to play a key role in the domain organization of the chromatin (*Conte et al., 2020*; *Shin and Brangwynne, 2017*) and in the regulation of genomic processes (*Hnisz et al., 2017*; *Kilic et al., 2019*; *Laghmach et al., 2021*; *Nozaki et al., 2017*; *Spegg and Altmeyer, 2021*; *Uchino et al., 2022*).

Some studies have reported that chromatin mobility is enhanced due to active transcription (*Gu et al., 2018*; *Tunnacliffe and Chubb, 2020*), whereas others report rather a decrease in mobility (*Mach et al., 2022*). Furthermore, other studies report diverse effects of RNA polymerase II inhibition on chromatin motion (*Germier et al., 2017*; *Ku et al., 2022*; *Shaban et al., 2018*). It has been also shown that the removal of RNA polymerase II from chromatin relaxes chromatin and increases its mobility (*Babokhov et al., 2020*). Conversely, there is an established view that chromatin mobility at the sites of double-stranded DNA breaks increases concomitant with their repair (*Eaton and Zidovska, 2020*; *Hauer and Gasser, 2017a*; *Hauer and Gasser, 2017a*; *Hauer et al., 2017b*; *Nagai et al., 2010*). Analysis of fluorescently tagged histones using displacement correlation spectroscopy has shown that chromatin undergoes coherent micron-scale motion at the time scales of 5–10 s independently of the cell cycle stage in mammalian cells (*Zidovska et al., 2013*). This coherent motion extended beyond individual chromosomes, suggesting mechanical coupling between chromosomes. Furthermore, the correlated motion of chromatin was ATP-dependent and completely disappeared upon DNA damage induction (*Eaton and Zidovska, 2020*; *Zidovska et al., 2013*).

DNA replication is a highly conserved energy-dependent process occurring in S-phase of the cell cycle, when chromatin structures undergo extensive reorganization to facilitate DNA synthesis (*Vincent et al., 2008*). An early study in budding yeast (*Heun et al., 2001*) demonstrated that individual heterologous loci became constrained in S-phase when integrated close to early- and late-firing replication origins, but not at the telomeric or centromeric regions. However, changes in chromatin mobility in S-phase were not observed when analyzing it at the level of chromosome territories in mammalian cells (*Walter et al., 2003*). Recent work using a CRISPR-based DNA imaging system suggests that local chromatin motion is restricted upon S-phase entry and more markedly in mid-late S-phase (*Ma et al., 2019*).

Altogether, it is not clear whether and how chromatin mobility changes during DNA replication and a mechanism behind the changes in chromatin motion. Therefore, it is important to address how changes in structure and metabolism of chromatin affect its mobility. It is quite intriguing to postulate that the process of genome duplication in mammals, which is performed at the level of naked DNA and involves local chromatin decondensation and rearrangements at the complete hierarchy of domains (*Baddeley et al., 2010*; *Chagin et al., 2019*; *Löb et al., 2016*; *Sadoni et al., 2004*; *Sporbert et al., 2002*), is associated with changes in chromatin mobility. Furthermore, it is tempting to speculate that the modulation of LCD may play a regulatory role; for example, by helping to define the transcriptional profile of the nucleus, by provoking collisions between regulatory regions, promoter regions, and transcription factories. These events could be halted or slowed down during the replication of the genome, avoiding collisions of the transcription with the replication machineries.

An alternative but not mutually exclusive model is that changes in LCD result from the execution of nuclear processes such as transcription or replication. This is particularly appealing as DNA/RNA helicases and polymerases are, in essence, motor proteins that reel DNA through. To distinguish between these possibilities, alterations in LCD must be characterized within the context of relevant nuclear processes and by labeling DNA directly and in an unbiased manner.

The process of genome replication has a particular and intrinsic connection between chromatin organization and the spatio-temporal progression of genome replication (reviewed in *Mamberti and Cardoso, 2020*). In that sense, firing of origins of replication by the activation of DNA helicase complexes followed by the loading of synthetic polymerase complexes tracks chromatin compaction and upon DNA duplication the focal chromatin organization at multiple hierarchical levels is preserved and can be detected over several cell generations (*Cremer et al., 2020*; *Jackson and Pombo, 1998*; *Sadoni et al., 2004*; *Sparvoli et al., 1994*). Importantly, genome replication is the only DNA metabolic process that encompasses the entire genome, thus ensuring the preservation of the genetic material upon cell division.

As most of the studies introduce artificial DNA sequences in genomic loci and use a large array of chromatin binding proteins to visualize the loci, chromatin dynamics may be altered in the subsequent process (*Germier et al., 2017*). Therefore, a more direct way to measure chromatin dynamics is to label and track the DNA directly (*Schermelleh et al., 2001*). A similar procedure has previously been used to mark chromosome territories and characterize their long-term rearrangements (*Bornfleth et al., 1999*; *Pliss et al., 2009*; *Walter et al., 2003*).

In this study, we investigated the mobility of chromatin in human cells, focusing on how changes in chromatin mobility are influenced by cell cycle progression and, in particular, DNA replication. To achieve this, we performed a detailed analysis of chromatin mobility in S-phase by combining locus-independent global labeling of DNA with reliable particle tracking. Measurement of the DNA content of the labeled structures allowed us to elucidate whether DNA replication affects chromatin mobility at the level of replication domains. Our results show that chromatin mobility generally decreases during S-phase and, in particular, at the proximity of the DNA polymerase complexes. Furthermore, we extended our study to dissect mechanisms behind the S-phase-related changes in chromatin mobility and inhibited DNA synthesis using small molecule inhibitors. We showed that chromatin mobility is further decreased in S-phase after inhibition of DNA synthesis. These results imply that loading of the polymerase complexes rather than the synthesis of DNA per se restraints DNA mobility.

## Results and discussion
### Genome-wide labeling of DNA and quantification of labeled chromatin domains

To evaluate LCD relative to the cell cycle stage, we first developed an experimental system to monitor both replication and chromatin changes in living cells in real time. We generated HeLa cell lines that stably express GFP-tagged proliferating cell nuclear antigen (PCNA) and single stranded DNA binding protein or replication protein A (RPA) (Materials and methods, *Supplementary file 1a*). We transfected fluorescent PCNA plasmid to label replication sites in human normal diploid fibroblasts (IMR90) (*Nichols et al., 1977*). PCNA is a core component of the DNA replication machinery and a marker for cell cycle progression (*Figure 1A*; *Chagin et al., 2016*; *Easwaran et al., 2005*; *Leonhardt et al., 2000*; *Moldovan et al., 2007*; *Prelich et al., 1987*). To visualize the mobility of native chromatin, we took the advantage of the ongoing DNA replication. We delivered a pulse of the fluorescently labeled nucleotide Cy3-dUTP by electroporation into an asynchronously growing population of human HeLa GFP-PCNA tumor cells and human diploid IMR90 fibroblasts, which allowed us to study chromatin dynamics in a global genome-wide manner (Materials and methods, *Figure 1A*). The nucleotide is incorporated into the nascent DNA of the cells in various periods of S-phase, effectively labeling the chromatin directly in an unbiased manner (*Manders et al., 1999*; *Sadoni et al., 2004*; *Schermelleh et al., 2001*).

The Cy3-dUTP-labeled chromatin structures were stable over the cell cycle progression and in subsequent cell cycles. Using time-lapse microscopy, we followed the cells that incorporated nucleotides in the initial S-phase stage over subsequent cell cycles. We used GFP-PCNA nuclear pattern to determine the cell cycle stages and sub-periods of S-phase (Materials and methods, Microscopy). This

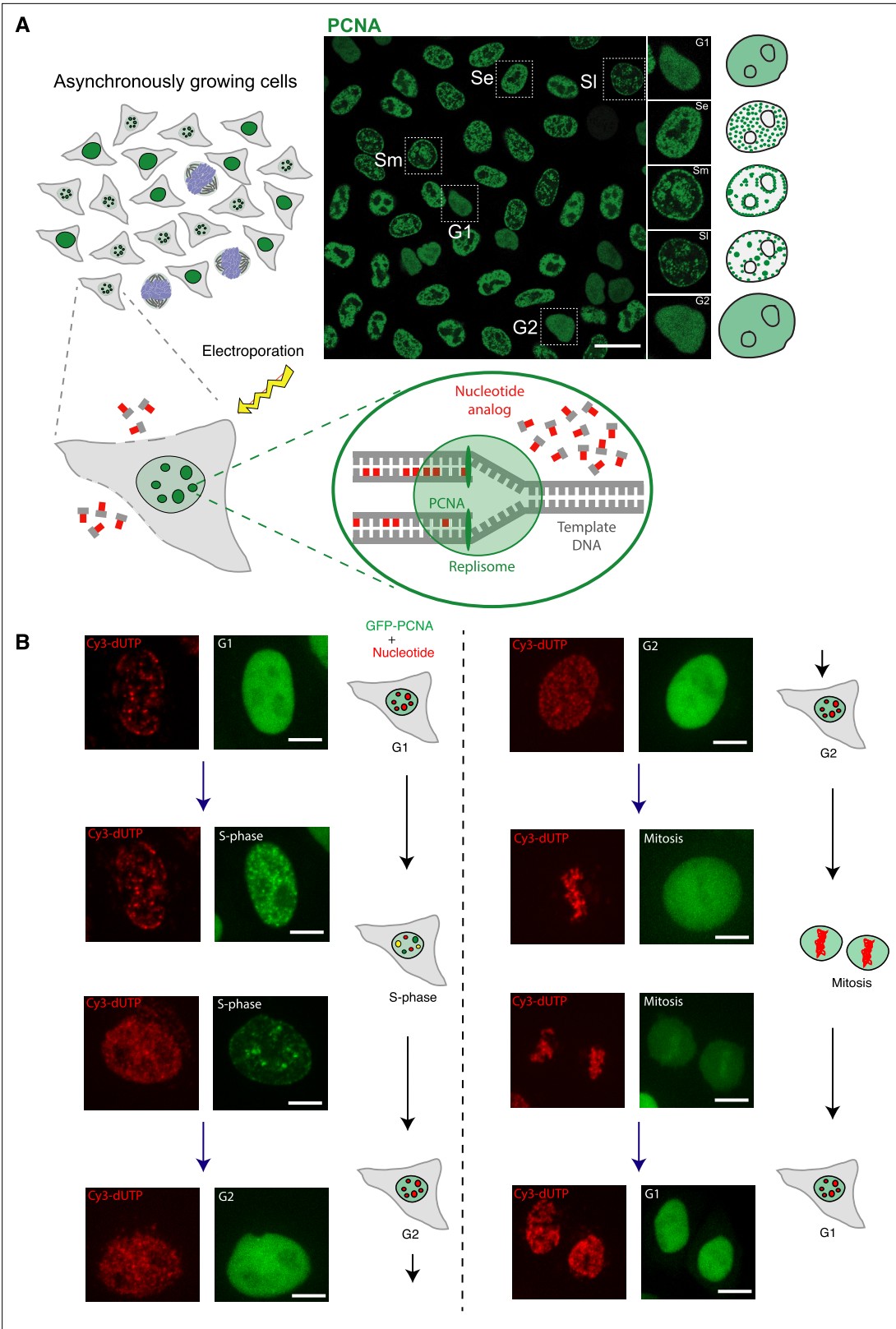

**Figure 1.** Incorporation of Cy3-dUTP in HeLa cell nuclei labels the whole genome randomly and with equal probability. (**A**) Schematic illustration of the labeling system for monitoring chromatin mobility and cell cycle progression. During S-phase, proliferating cell nuclear antigen (PCNA) accumulates within the nucleus at sites of active DNA replication and exhibits a distinct puncta pattern. During G1 and G2, GFP-PCNA is diffusely distributed throughout the nucleus. Asynchronously growing populations of cells were exposed to electroporation to promote the uptake of Cy3-dUTP. In cells

*Figure 1 continued on next page*

*Figure 1 continued*

undergoing DNA replication, this fluorescent nucleotide is incorporated into nascent DNA strands at sites of active DNA replication, resulting in the direct fluorescent labeling of genomic segments. Based on the PCNA pattern, different cell cycle stages can be differentiated as shown in the image on the right (Se – early S, Sm – mid S, SL – late S, G1/G2 – gap phases, green – PCNA). (**B**) After Cy3-dUTP labeling (shown in red), cells were followed by time-lapse microscopy to identify the cell cycle (sub)stages and their progression. The representative images of different cells using time-lapse microscopy were shown to depict the patterns of PCNA (shown in green) in each sub-stage and their change over time (co-localized signals in yellow). This was used to classify cells in G1, S, and G2 phases of the cell cycle for motion analysis. Approximately 18–24 hr after nucleotide electroporation, Cy3-dUTP-labeled cells were imaged for motion analysis (see also *Videos 1–5*). The contrast of the images was adjusted linearly for visualization purposes. Scale bar: 5 µm.

allowed us to classify cells in different cell cycle stages and sub-periods of S-phase (G1, early S, mid S, late S, G2), which is illustrated in *Figure 1B* (see also *Videos 1–5*).

With this approach, DNA labeled during the pulse of Cy3-dUTP nucleotide corresponds to genomic regions replicated concomitantly during an S-phase sub-period. Since LCD measurements depend on the object size, it was important to evaluate the size of the labeled DNA domains. This allowed us to correlate the chromatin domain sizes and their diffusion rates. For this purpose, we measured the total DNA amount in a cell and the fraction of it that corresponded to the labeled domain (Materials and methods, DNA quantification of labeled chromatin). First, we applied chemical fixation to cells labeled with Cy3-dUTP using formaldehyde. The total DNA was then labeled using the DNA dye DAPI. Next, we segmented the entire nucleus as well as the individual labeled chromatin foci within the same cell. The fraction of DAPI intensity within the segmented replication focus ($I_{RFi}$) over the total DNA intensity within the cell ($I_{DNA\ total}$) yields the amount of DNA present per labeled chromatin focus (*Figure 2A*, *Figure 2—figure supplement 1*). Since nuclear DNA amount doubles continuously throughout the S-phase (*Chagin et al., 2016*; *Leonhardt et al., 2000*, *Figure 2B*), it was important to scale the total DNA amount by a correction factor depending on S-phase sub-stage to measure the DNA amount per focus more accurately. The relative amount of DNA throughout the cell cycle stages and sub-stages of S-phase was calculated and plotted as histograms, with the mean of the histogram for each cell cycle (sub)stage constituting the cell cycle correction factor (*Figure 2B*). The fraction of DAPI intensities were corrected by multiplication with the genome size corresponding to the cell cycle stage. The genome size of HeLa Kyoto cells is GS = 9.682 ± 0.002 Gbp (*Chagin et al., 2016*) and for IMR90 fibroblasts the genome size is 6.37 Gbp as measured earlier (*Nichols et al., 1977*). We plotted the DNA amount present in each replication focus in *Figure 2C* for HeLa on left and IMR90 on right. The highest frequency of average DNA amount per focus (mode +1 bin) was about 300–600 kbp of DNA (*Figure 2C*). Altogether, with our labeling approach, we labeled DNA domains of sizes ranging from 0.5 Mbp to 10 Mbp, with the vast majority corresponding to 0.5 Mbp, which correspond well to multi-loop chromatin domains corresponding in size to topological associated domains (reviewed in *Giorgetti and Heard, 2016*).

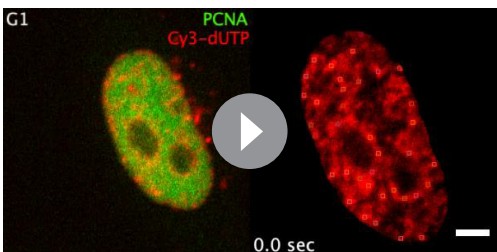

**Video 1.** Time-lapse microscopy of HeLa K cells in G1 phase expressing fluorescent proliferating cell nuclear antigen (PCNA) (green) and labeled chromatin (red). Scale bar: 5 µm.

https://elifesciences.org/articles/87572/figures#video1

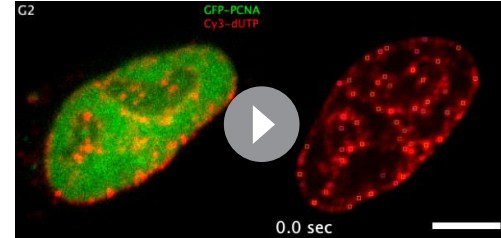

**Video 2.** Time-lapse microscopy of HeLa K cells in G2 phase expressing fluorescent proliferating cell nuclear antigen (PCNA) (green) and labeled chromatin (red). Scale bar: 5 µm.

https://elifesciences.org/articles/87572/figures#video2

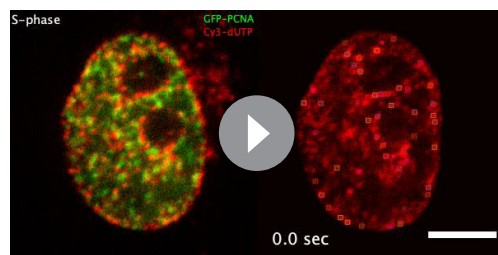

**Video 3.** Time-lapse microscopy of HeLa K cells in S-phase expressing fluorescent proliferating cell nuclear antigen (PCNA) (green) and labeled chromatin (red). Scale bar: 5 µm.

https://elifesciences.org/articles/87572/figures#video3

# Chromatin motion decreases in the S-phase of the cell cycle relative to the G1 and G2 phases

To determine how the global dynamics of chromatin changes during cell cycle progression, we used LCD measurements relative to the cell cycle stage. Live cell time-lapse image sequences of HeLa and IMR90 cells after labeling chromatin with Cy3-dUTP were obtained and motion analysis was performed to determine the type of motion (*Figure 3A*, Materials and methods). Normal diffusion or Brownian motion is a linear diffusion model with $\alpha=1$ and when $\alpha>1$ it is termed super diffusion. First, the cells were annotated according to the different cell cycle stages (G1, S, G2) based on the PCNA subnuclear pattern (Materials and methods). PCNA forms puncta or foci at the active replication sites during S-phase and this was used to classify cells in S-phase. We were able to distinguish between G1 and G2 cells, even though they exhibit a similar diffused PCNA subnuclear distribution, based on the information on the preceding cell cycle stage from the time-lapse analysis performed after Cy3-dUTP labeling (Materials and methods, Microscopy). Specifically, cells with diffusely distributed PCNA signal which had previously undergone mitosis were in G1 phase, whereas the ones with similar diffuse PCNA pattern that had previously undergone S-phase (punctated PCNA pattern) were classified as being in G2 phase (*Figure 1B*). The PCNA signal was also used to segment the nucleus, and the individual chromatin foci were detected within the segmented nuclei. Probabilistic tracking was performed to obtain individual chromatin trajectories (*Figure 3B*; *Figure 3—figure supplement 1*). In case of IMR90 cells, affine image registration was performed using the method in *Celikay et al., 2022*, to address the stronger cell movement compared to HeLa cells. This was followed by a mean square displacement (MSD) analysis to determine the chromatin motion in different cells (*Figure 3—figure supplement 1*). In fixed cells, labeled chromatin foci showed almost no motion, which was used as a control for the stability of the imaging system and the tracking protocol. We plotted the MSD over time (up to 20 s) for chromatin foci in cells from different cell cycle stages as well as for fixed cells (*Figure 3C*). As we focused on chromatin mobility changes during S-phase, the G1, G2 cells were together in *Figure 3C*. The MSD curves of G1, G2, S-phase (separated) are plotted in *Figure 3—figure supplement 1B*.

We observed significantly constrained global chromatin motion in S-phase cells compared to non-replicating G1/G2 cells suggesting that chromatin was more constrained during DNA replication. This effect was stronger in IMR90 cells compared to HeLa Kyoto. The table shows the average diffusion rates (*Figure 3*). For HeLa average diffusion rate of chromatin in G1/G2 was $D=133.6$ µm²/s $\times$ $10^{-5}$, whereas the diffusion rates dropped to $D=105.6$ µm²/s $\times$ $10^{-5}$ during S-phase (*Figure 3C*). For IMR90 average diffusion rate of chromatin in G1/G2 was $D=86.5$ µm²/s $\times$ $10^{-5}$, whereas the diffusion rates dropped to $D=43.7$ µm²/s $\times$ $10^{-5}$ during S-phase

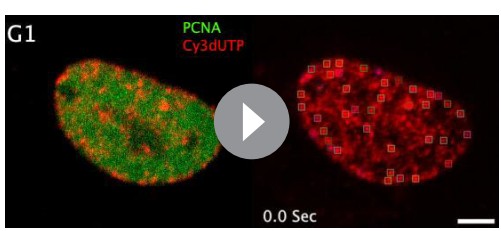

**Video 4.** Time-lapse microscopy of IMR90 cells in G1 phase expressing fluorescent proliferating cell nuclear antigen (PCNA) (green) and labeled chromatin (red). Scale bar: 5 µm.

https://elifesciences.org/articles/87572/figures#video4

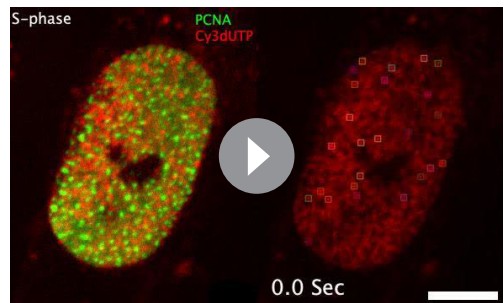

**Video 5.** Time-lapse microscopy of IMR90 cells in S-phase expressing fluorescent proliferating cell nuclear antigen (PCNA) (green) and labeled chromatin (red). Scale bar: 5 µm.

https://elifesciences.org/articles/87572/figures#video5

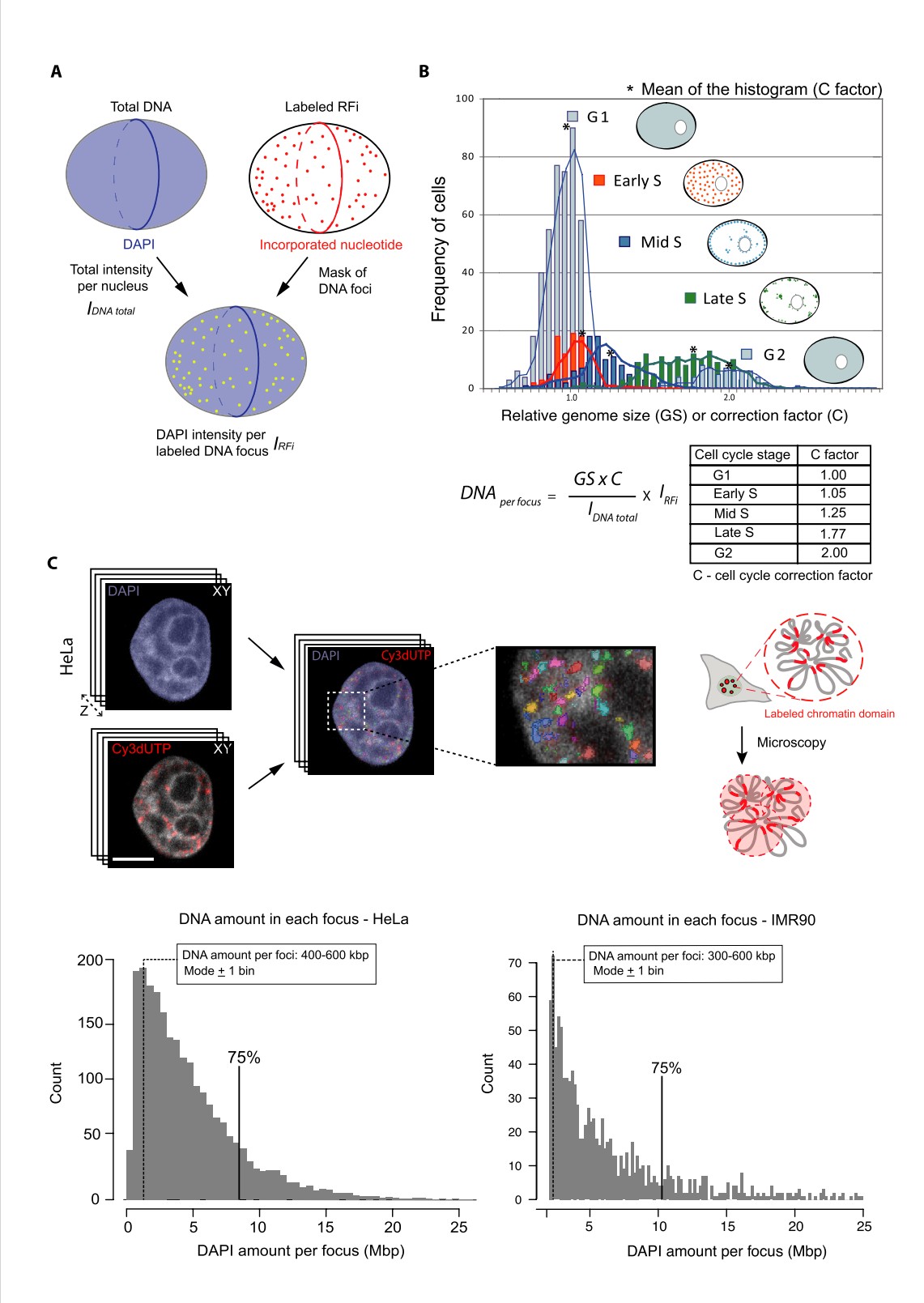

The equation shown in the figure:

$$DNA_{per\ focus} = \frac{GS \times C}{I_{DNA\ total}} \times I_{RFi}$$

| Cell cycle stage | C factor |
|---|---|
| G1 | 1.00 |
| Early S | 1.05 |
| Mid S | 1.25 |
| Late S | 1.77 |
| G2 | 2.00 |

C - cell cycle correction factor

**Figure 2.** Principle of measuring DNA content per labeled DNA focus using confocal data. (**A**) To determine the amount of DNA per labeled DNA focus, we used the total DAPI signal (DNA amount) of the segmented whole nucleus ($I_{DNA\ TOTAL}$). The DNA intensity per labeled DNA focus within the segmented foci is obtained ($I_{RFi}$) by masking the replication foci and estimating the corresponding portion of DAPI signal. (**B**) Throughout the S-phase progression the amount of DNA increases twofold from early to late S-phase. The amount of DNA present in the nucleus at a particular cell cycle

*Figure 2 continued on next page*

*Figure 2 continued*

stage can be determined by measuring the DNA amount in the population of cells, while using the proliferating cell nuclear antigen (PCNA) pattern to determine the cell cycle stage and S-phase sub-stage (see also *Figure 1*). The relative mean amount of DNA of each of the cell cycle (sub-)stages is used to calculate the cell cycle correction factor. The cell cycle correction factor (C/cell cycle stage) was estimated as: 1.0/G1; 1.05/early S-phase; 1.25/ mid-S-phase; 1.77/late S-phase, 2/G2. The G1 genome size (GS) for HeLa cells is 9.7 Gbp (*Chagin et al., 2016*). The amount of DNA per labeled focus is the ratio of $I_{RFi}$ and $I_{DNA\ TOTAL}$ multiplied by C × GS. (**C**) The illustration on right depicts the imaging of labeled replication foci using confocal microscopy. DNA quantification of replication labeled foci in tumor HeLa and normal diploid IMR90 cells was done by imaging full Z-stacks volume of chromatin labeled with Cy3-dUTP and DNA with DAPI and imaged using confocal spinning disk microscopy (*Figure 2—figure supplement 1*, *Supplementary file 1e*). The histogram represents the DNA amount per focus for labeled S-phase cells (N=30 cells) for HeLa and IMR90 cells. The mode ±1 bin of the histogram represents the highest frequency of average size of replication domains labeled (300–600 kbp). Scale bar: 5 μm.

The online version of this article includes the following figure supplement(s) for figure 2:

**Figure supplement 1.** Pipeline and controls for DNA quantification of labeled replication foci using confocal data.

(*Figure 3C*). We computed the α values in different stages, which define the type of diffusion motion. Chromatin exhibited anomalous subdiffusion or obstructed diffusion with $0.1 < α < 0.9$. Anomalous diffusion of cellular structures including chromatin with α values between 0.1 and 0.9 have been reported (*Bronshtein et al., 2015*; *Ghosh and Webb, 1994*; *Mach et al., 2022*; *Oliveira et al., 2021*; *Simson et al., 1998*; *Smith et al., 1999*).

In agreement with our results, it has been initially reported in yeast that some chromatin loci are constrained during S-phase (*Heun et al., 2001*). This study has been extended to the mammalian genome using the CRISPR targeted labeling of specific genomic loci to demonstrate that the S-phase mobility of the labeled chromosomal loci decreases in S-phase compared to G1/G2 (*Ma et al., 2019*). Another study reported that during DNA replication there were changes in chromatin mobility due to an unknown mechanism (*Nozaki et al., 2017*).

As we measured, decrease in global chromatin motion during S-phase, which includes labeled chromatin, which is replicating as well as non-replicating, we next focused the study on the microenvironment of active replication sites. This opened the question of whether the loading of the replisome on chromatin or its enzymatic activity during S-phase actively restricted chromatin motion. Hence, we analyzed in detail the spatial relationship of chromatin diffusion and DNA replication sites.

## Chromatin motion decreases in proximity to active DNA replication sites

DNA replication involves systematic and structured assembly of proteins directly or indirectly involved in DNA synthesis. DNA replication factors such as DNA polymerase clamp protein (PCNA), the DNA helicase complex that unwinds DNA, and the replication protein A (RPA) complex, which stabilizes and protects the ssDNA exposed upon helicase activity are illustrated in *Figure 4A*. The DNA polymerase clamp PCNA, one of the most well-studied replication proteins, was used to mark the active DNA replication sites. To test whether DNA replication factors restrict chromatin motion, we performed proximity analysis (Materials and methods, *Figure 4B*). As before, we used Cy3-dUTP to label chromatin in the S-phase of the previous cell cycle. We then followed the cells through the cell cycle to select cells in which some of the sites of labeled chromatin were replicating in the S-phase of the next cell cycle at the time of observation. This allowed us to image the labeled chromatin marked in the previous cell cycle together with a live cell marker (fluorescent PCNA) for the active replication sites in the next cell cycle (*Figure 4B*). Subsequently, we measured the mobility of chromatin from these S-phase cells at increasing center-to-center distances (CCD, R) from active replication sites (*Figure 4C*). For chromatin outside the CCD with replication sites in these S-phase cells, we observed the same diffusion rate as before for the chromatin foci in S-phase cells with no differentiation of whether chromatin was actively replicating or not (*Figure 3*). However, we observed that the chromatin in the proximity of replication sites (actively replicating) had more restricted motion when located up to 1 μm (center to center) distance to an active replisome, and this effect vanished at higher distances (*Figure 4C*, *Figure 4—figure supplement 1*). These data indicate that the reduction of chromatin motion in S-phase is spatially correlated with DNA replication and suggest that DNA synthesis restricts chromatin motion. Hence, we next investigated whether loading of the DNA replication machinery restricts chromatin motion or alternatively DNA synthesis activity is responsible for it.

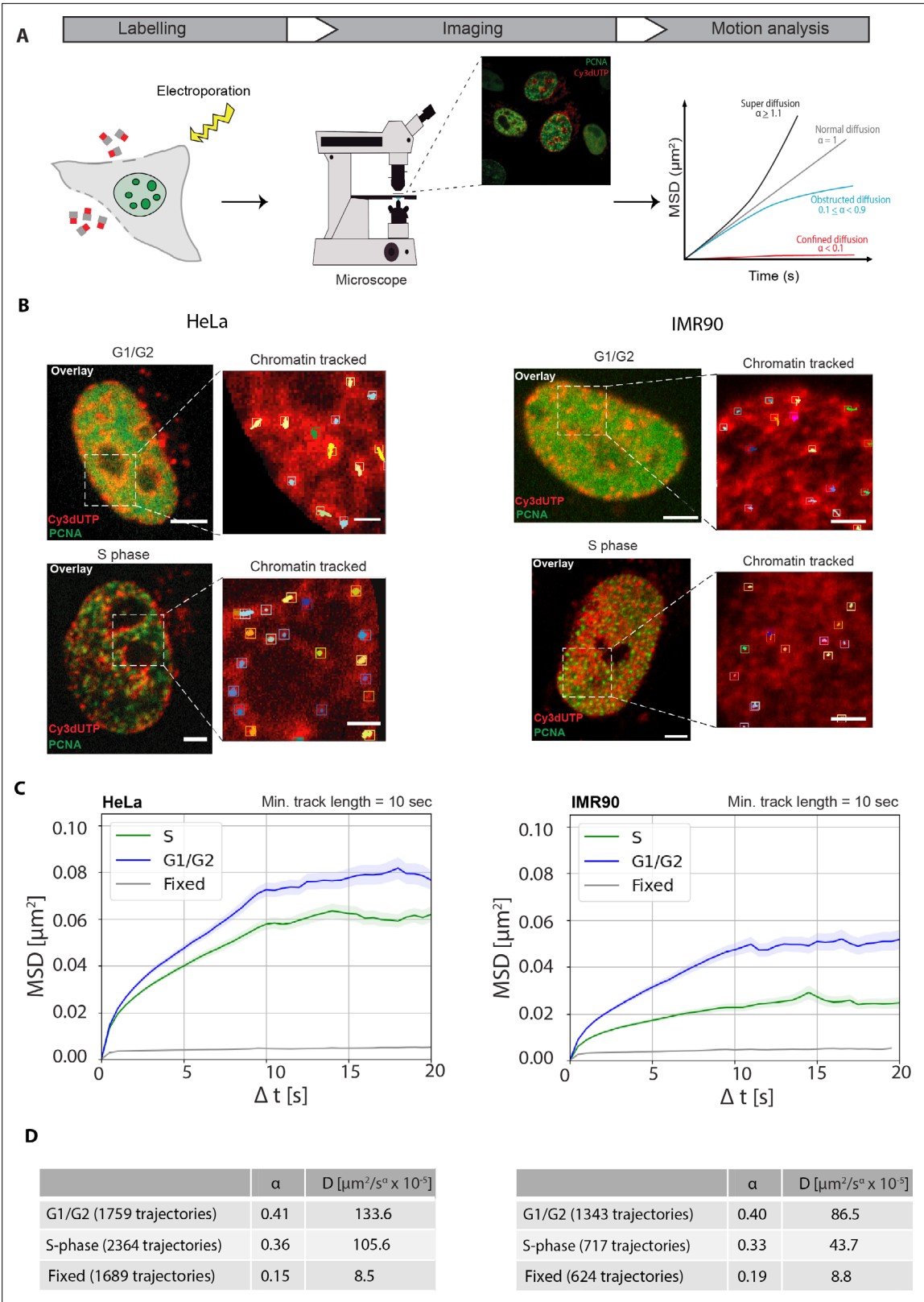

**Figure 3.** Single-particle motion analysis of labeled chromatin throughout the cell cycle. (**A**) Brief schematics of the main steps of motion analysis starting with chromatin labeling using fluorescently labeled nucleotides via electroporation, followed the next day (i.e., cell cycle) by confocal time-lapse imaging of the chromatin channel and performing motion analysis on computed tracks to determine the diffusion rates of chromatin (Materials and methods). (**B**) Overlay images of HeLa Kyoto and IMR90 cells expressing GFP/miRFP proliferating cell nuclear antigen (PCNA) and labeled chromatin

*Figure 3 continued on next page*

*Figure 3 continued*

(Cy3dUTP) in different cell cycle stages (G1/G2 – diffused PCNA, S-phase – PCNA puncta). Cropped region (white box) showing the chromatin tracks of individual foci in both G1/G2 and S-phase cells. The aggregates of Cy3-dUTP that are found in cytoplasm are excluded from the analysis using a nuclear mask. See also *Videos 1–5*. (**C**) Result of motion analysis of computed chromatin tracks for different cell cycle stages (*Figure 3—figure supplement 1*) for HeLa and IMR90 cells. Mean square displacement (MSD, µm²) curves were plotted over time (s). MSD curves for G1/G2, S-phase, fixed cells with a minimum track length of 10 s, and a total time of 20 s were plotted with error bars (SEM – standard error of the mean) representing the deviations between the MSD curves for an image sequence in transparent color around the curve. Scale bar: 5 µm. Insets scale bar: 1 µm. (**D**) The tables provide the detailed information on the number of trajectories per condition along with average diffusion rates and anomalous α coefficient showing subdiffusion.

The online version of this article includes the following figure supplement(s) for figure 3:

**Figure supplement 1.** Workflow of chromatin motion and proximity analysis for confocal data.

## DNA synthesis inhibition leads to activation of DNA helicases and accumulation of ssDNA binding proteins and DNA polymerases

During DNA replication, replisome components are assembled at the origin of replication to form an active replisome (*Casas-Delucchi and Cardoso, 2011*; *Yao and O'Donnell, 2010*; *Yao and O'Donnell, 2016*). To test whether the process of DNA synthesis itself is responsible for constraining chromatin, we analyzed chromatin motion after inducing replication stress. By treating cells with aphidicolin, DNA synthesis is slowed down or stopped altogether (*Vesela et al., 2017*). Aphidicolin is a tetracyclic antibiotic isolated from *Nigrospora sphaerica*, which interferes with DNA replication directly by inhibiting DNA polymerases α, ε, and δ (*Bambara and Jessee, 1991*; *Baranovskiy et al., 2014*; *Byrnes, 1984*; *Cheng and Kuchta, 1993*). Our hypothesis was that it is the loading of replisome components that affects the chromatin motion (LCD). Therefore, we focused on LCD measurements after inhibiting DNA synthesis directly with aphidicolin and characterized the effects on chromatin motion in order to understand the mechanism behind it.

First, we tested in detail the rate and level of inhibition of DNA synthesis with aphidicolin (150 µM) using thymidine analogs (in this case EdU), which get incorporated into newly synthesized DNA and can be detected using click-IT chemistry (Materials and methods). We visualized GFP-PCNA and EdU in fixed cells and performed high-throughput image analysis to characterize the effect of aphidicolin on DNA synthesis inhibition at different time points (Materials and methods, *Figure 3—figure supplement 1*). We observed that DNA synthesis was inhibited minutes after aphidicolin treatment. Using high-content microscopy, we quantified the population of cells actively synthesizing DNA (EdU signal) upon stress and observed that in almost 99% of the cell population, DNA replication was inhibited within half an hour of aphidicolin incubation (Materials and methods, *Figure 5A*, *Figure 5—figure supplement 1*, *Figure 5—figure supplement 2*). Subsequent experiments were all performed with these conditions.

Secondly, we made use of the above conditions in which DNA synthesis was inhibited and analyzed the consequences of replication stress on the replisome components and their kinetics. For this purpose, we performed time-lapse microscopy of GFP-PCNA and GFP-RPA34 expressing cells. During active DNA synthesis, the DNA polymerase clamp and processivity factor PCNA is loaded onto the DNA as a trimeric ring and is tightly bound to the DNA (*Figure 5A*). During aphidicolin treatment though, PCNA dissociated from DNA as shown before (*Görisch et al., 2008*; *Rausch et al., 2021*; *Figure 5A*). Aphidicolin treatment does not stop helicase activity and the replication protein A (RPA) is loaded on the ssDNA after being unwound by the DNA helicase. The more the DNA double helix is unwound, the more RPA loads onto the ssDNA generated (*Rausch et al., 2021*). For this analysis, we generated a HeLa cell line stably expressing GFP-RPA34 (*Figure 5—figure supplement 3*). We performed time-lapse microscopy on HeLa GFP-RPA34 cells every 5 min for 60 min for both aphidicolin-treated and control DMSO-treated cells (*Figure 5—figure supplement 4A*). We observed that RPA accumulated over time on DNA at replication sites in aphidicolin-treated cells but not in the control cells (*Figure 5—figure supplement 4*). RPA accumulation indicated that the DNA helicase complexes continued unwinding the DNA, which allowed for increasing amounts of RPA to bind and, at the same time, the DNA polymerases were not active displacing the RPA while synthesizing the second (complementary) DNA strand (*Görisch et al., 2008*). Therefore, we studied the kinetics of accumulation of RPA on chromatin upon DNA synthesis inhibition by quantifying the accumulation of GFP-RPA34 in live cells upon treatment with aphidicolin normalized to DMSO-treated cells using the

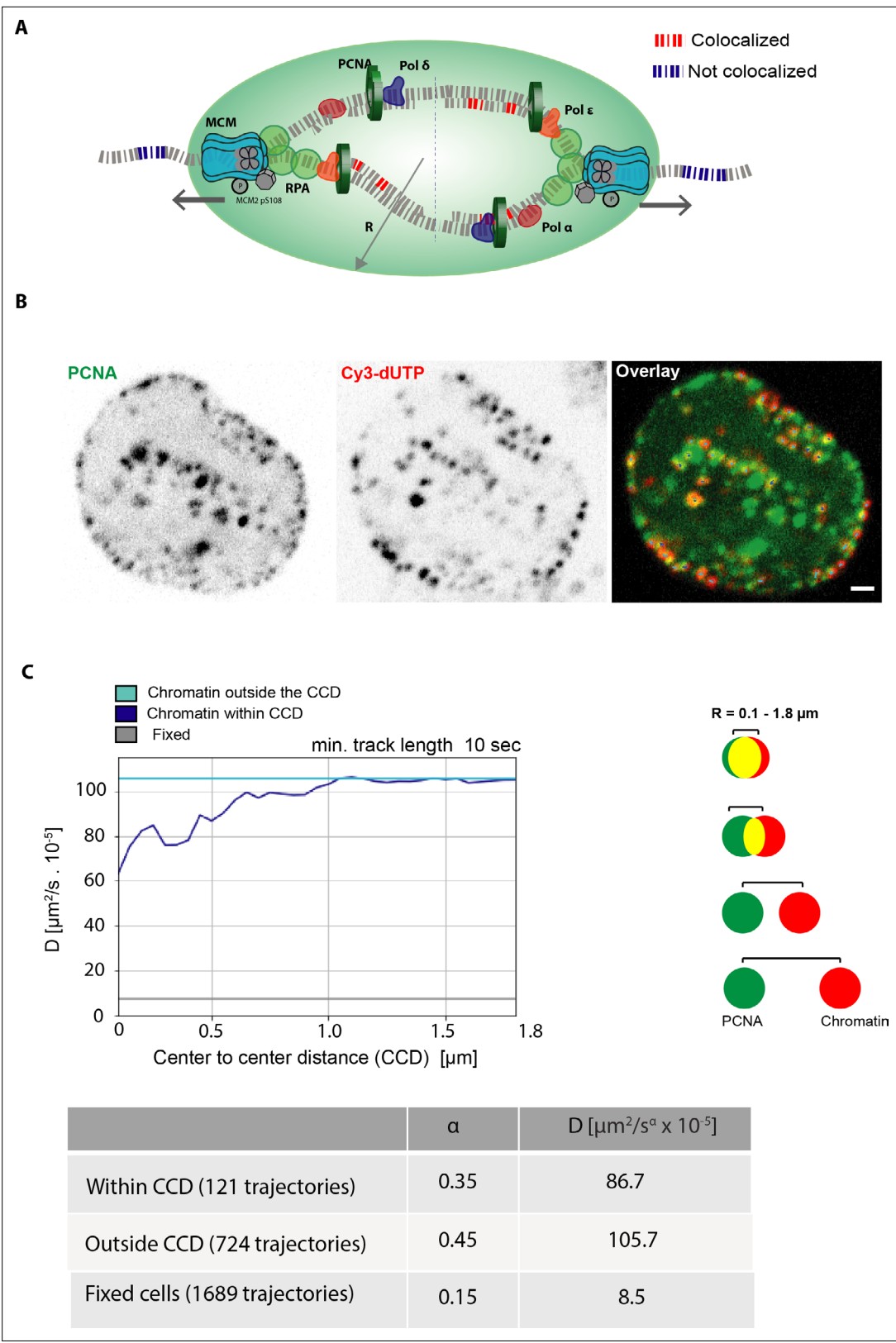

**Figure 4.** Analysis of chromatin mobility versus distance (proximity) to the DNA replication machinery. (**A**) Schematic illustration of replisome components (helicase, replication protein A, proliferating cell nuclear antigen [PCNA]) actively replicating chromatin. The geometric centers of the labeled chromatin foci and labeled replication sites were first defined. The chromatin within the defined center-to-center distance (CCD) to a PCNA-labeled replication site is defined as chromatin that is within CCD and, otherwise, is defined as outside the CCD. (**B**) In order to obtain mobility

*Figure 4 continued*

information of labeled chromatin in the proximity of PCNA foci (active replication sites) one frame of PCNA channel was acquired followed by 50 frames of the chromatin channel with a frame rate of 0.5 s. The images show the spatial distribution of PCNA and chromatin foci (Cy3-dUTP). (**C**) The graph represents the average diffusion rates of the mean square displacement curves (MSD) of chromatin within the CCD and chromatin outside the CCD with increasing distance (R) measured between the centers of PCNA and chromatin foci (*Figure 4—figure supplement 1*).The table below provides the detailed information on number of trajectories per individual sample along with average diffusion rates (µm²/s × 10⁻⁵) and anomalous α coefficient showing subdiffusion at 0.5 µm CCD. Scale bar: 1 µm.

The online version of this article includes the following figure supplement(s) for figure 4:

**Figure supplement 1.** Mean square displacement (MSD) analysis of labeled chromatin in the proximity of active replication sites at varying center-to-center distances (CCD).

coefficient of variation (Cv), which indicates the amount of RPA protein accumulated over time (Materials and methods, *Figure 5B*, *Figure 5—figure supplement 5*). We observed clear accumulation of RPA over time relative to control, showing that the ssDNA binding protein accumulates on chromatin. Hence, this indicates that upon stress the DNA helicase remained active unwinding the DNA.

Next, we analyzed the distribution of the helicase subunit MCM2 and its phosphorylated (p108) form along with DNA polymerases α, ε, and δ (*Supplementary file 1d*) at the chromatin. It has been previously described that the phosphorylated form of MCM2 is the active form for DNA unwinding (*Forsburg, 2004*; *Montagnoli et al., 2006*). We predicted from the RPA accumulation that the helicase subunit was present at the replication sites and actively spooling the DNA through after the DNA synthesis inhibition. We first performed western blot analysis of different replication factors from asynchronous populations of HeLa cells after isolating the cytoplasm, nucleoplasm, and chromatin fractions (Materials and methods). We tested the fractionation protocol by blotting the membranes with antibodies to α-tubulin for the cytoplasmic fraction and macro H2A1 histone for the chromatin fraction (*Figure 5C*). The same fractions were then incubated with antibodies for different replication factors. We observed significant dissociation of PCNA from chromatin and accumulation of RPA on chromatin upon aphidicolin treatment (*Figure 5C*) consistent with our fixed cell and live cell microscopy analysis. We found no significant changes in MCM2 helicase subunit levels on chromatin and higher levels of phosphorylation of MCM2 upon treatment with aphidicolin (*Figure 5C*). Lastly, we incubated the blots with antibodies recognizing the catalytic subunits of the DNA polymerases α, δ, and ε complexes (Materials and methods, *Supplementary file 1d*). The DNA polymerases showed a different behavior as compared to the DNA polymerase clamp protein, with DNA polymerase α being enriched on chromatin upon stress, with only minor to no changes being observed for the processive DNA polymerases δ and ε (*Figure 5C and D*). It is of note that both these processive DNA polymerases bind the polymerase clamp PCNA whereas the far less processive DNA polymerase α does not. The full-length blots are shown in *Figure 5—figure supplement 6*.

We then performed an orthogonal analysis using high-throughput microscopy and image analysis. We labeled cells with EdU for 10 min to mark the S-phase cells and treated cells with DMSO/aphidicolin and subsequently performed pre-extraction to remove the unbound fraction of proteins and only detect the chromatin-bound proteins. In this manner, we separately quantified accumulation only in S-phase cells and not in populations of cells including all cell cycle stages as in the previous western blot analysis (*Figure 5C*). The pre-extracted cells were fixed and immunostained for different replisome components (*Figure 5D*). The cells were imaged using a spinning disk confocal microscopy system (*Supplementary file 1e*) and image analysis was performed using the KNIME software with custom pipeline to quantify the accumulation/loss of replication factors on chromatin in S-phase cells (Materials and methods, *Supplementary file 1i*, *Figure 5—figure supplement 7*, *Figure 5—figure supplement 8*). Using the EdU signal, the S-phase cells were selected for the quantitation of chromatin-bound replisome components (*Figure 5D*). We found that PCNA dissociated from chromatin and RPA accumulated on chromatin upon stress in accordance with our previous analysis (*Figure 5C*). We found no changes in MCM2 helicase subunit but an increase in active MCM2p108 levels upon stress. This is consistent with no new loading of DNA helicases but de novo activation of already loaded helicase complexes (*Ge et al., 2007*; *Ibarra et al., 2008*). Finally, we observed significant accumulation of DNA polymerases α and δ on chromatin after aphidicolin treatment. PCNA does not associate with DNA polymerase α, which has a low processivity, but it associates with DNA polymerases ε and δ, which constitute the processive synthetic machinery responsible for most of the duplication of the genome.

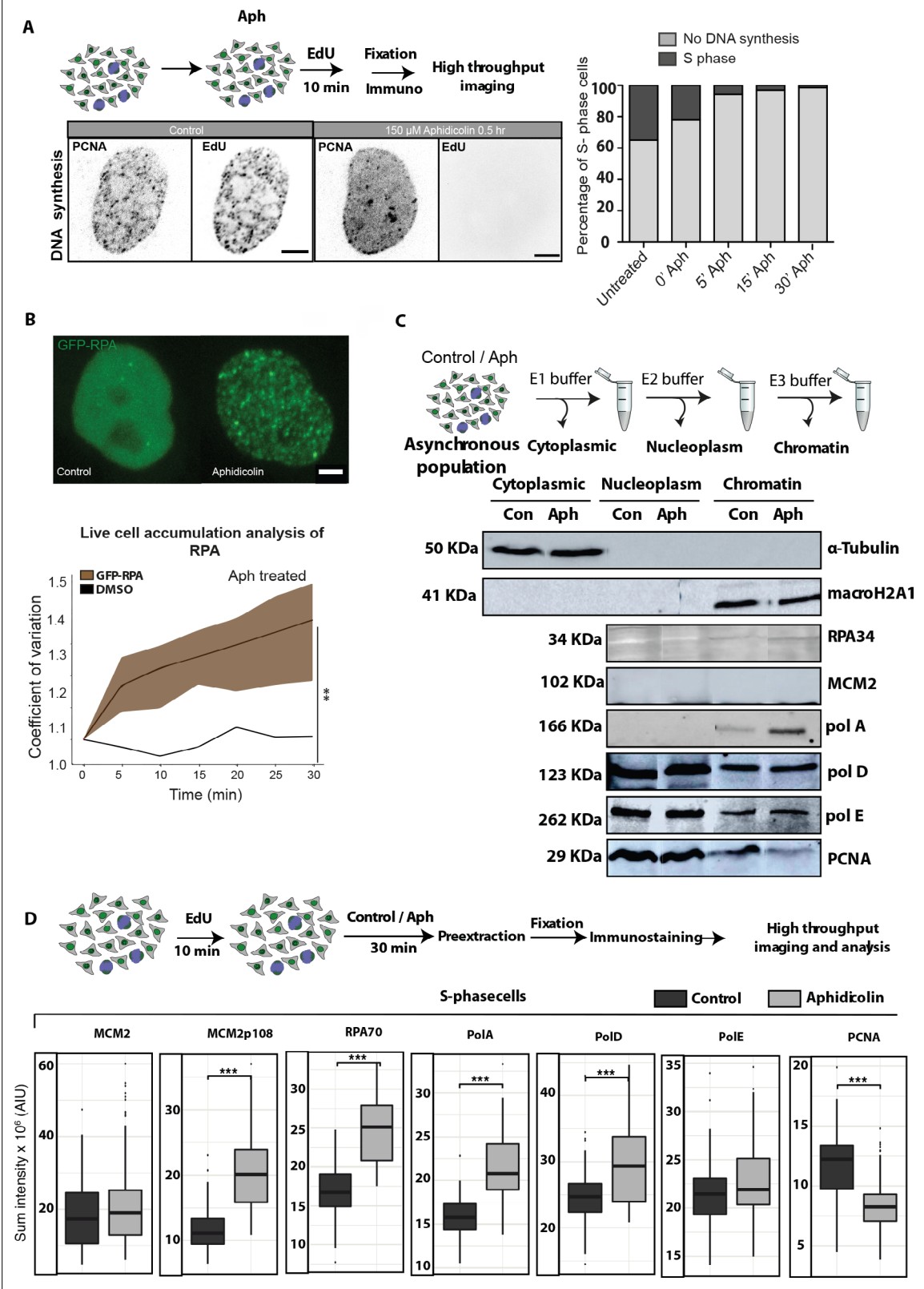

**Figure 5.** Dissecting the kinetics of replisome components after inhibition of DNA synthesis. (**A**) The use of thymidine nucleotide analogs like 5-ethynyl-2'-deoxyuridine (EdU), which is incorporated into replicating DNA, allows us to estimate the time needed for complete inhibition of DNA synthesis. The representative images show no incorporation of EdU in S-phase cells upon aphidicolin (Aph) treatment for 30 min. The plots below the images depict the % of cells with no DNA synthesis as scored by the EdU signal and the corresponding % of cells still replicating DNA (*Figure 5—figure supplement*

*Figure 5 continued on next page*

*Figure 5 continued*

*1, Figure 5—figure supplement 2*). (**B**) The line plot shows the live cell accumulation analysis showing the normalized average RPA70 accumulation at replication sites (coefficient of variation ± standard deviation in transparent color) of HeLa cells stably expressing GFP-RPA34 (*Figure 5—figure supplements 3–5*). (**C**) Western blots of cytoplasm, nucleoplasm, and chromatin fractions of asynchronous population of HeLa cells probed for different replication factors. The western blots shown are cropped from the same replicates for easier visualization without contrast adjustment and the full blots are shown and highlighted in *Figure 5—figure supplement 6*. (**D**) HeLa cells were pulsed with EdU for 10 min to identify S-phase cells and pre-extracted to detect chromatin-bound proteins and different replication factors were detected using immunofluorescence. High-throughput imaging and image analysis were performed (*Figure 5—figure supplements 7 and 8*). Box plots depict the accumulation of the replisome factors indicated at DNA replication sites. Same Y-axis scale plots are shown in *Figure 5—figure supplement 8*. The box plot lower and upper hinges correspond to the first and third quartiles (the 25th and 75th percentiles), the upper whisker extends from the hinge to the largest value no further than 1.5× IQR from the hinge (where IQR is the interquartile range, or distance between the first and third quartiles). The lower whisker extends from the hinge to the smallest value at most 1.5× IQR of the hinge. The horizontal line represents the median value. The outliers plotted individually as separate dots outside of the whiskers. ***p<0.001 by Wilcoxon rank-sum test, for aphidicolin-treated versus control sample. Scale bar: 5 µm.

The online version of this article includes the following source data and figure supplement(s) for figure 5:

**Source data 1.** The original full image files of western blots in *Figure 5C*.

**Source data 2.** The original full image files of western blots in *Figure 5C*.

**Source data 3.** The original full image files of western blots in *Figure 5C*.

**Source data 4.** The original full image files of western blots in *Figure 5C*.

**Source data 5.** The original full image files of western blots in *Figure 5C*.

**Source data 6.** The original full image files of western blots in *Figure 5C*.

**Source data 7.** The original full image files of western blots in *Figure 5C*.

**Source data 8.** The original full image files of western blots in *Figure 5C*.

**Figure supplement 1.** Time course analysis of inhibition of DNA synthesis upon aphidicolin treatment.

**Figure supplement 2.** Schematic overview of the high-throughput analysis pipeline of DNA synthesis inhibition by aphidicolin.

**Figure supplement 3.** HeLa GFP-RPA cell line characterization after generation using the Flp-recombinase system.

**Figure supplement 4.** Live cell time-lapse microscopy of HeLa GFP-RPA34 cells to determine RPA accumulation at replication sites upon aphidicolin treatment.

**Figure supplement 5.** Pipeline for the analysis of RPA enrichment at replication sites in living cells.

**Figure supplement 6.** Full-length western blots probed for different replication factors with cytoplasm, nucleoplasm, and chromatin fractions.

**Figure supplement 7.** Analysis of replisome component enrichment at replication sites upon replication stress.

**Figure supplement 8.** Pipeline for the analysis of replisome components on chromatin upon stress.

Hence, it was surprising that these two polymerases remain associated and even load de novo at non-synthetizing replication sites. The increase in DNA polymerase α could lead to the recruitment of alternative polymerase clamp 9-1-1 as reported before (*Michael et al., 2000*; *Van et al., 2010*; *Yan and Michael, 2009a*; *Yan and Michael, 2009b*). Several scenarios explaining the different levels of DNA polymerases α and δ upon stress are possible (*Figure 6—figure supplement 1*): (i) multiple polymerase complexes may load within the same Okazaki fragment, which is less likely in view of what is known on DNA replication (*Figure 6—figure supplement 1A*); (ii) multiple Okazaki fragments each with DNA polymerase α and δ within the same replication fork may form on the extended ssDNA unwound by the helicase complex (*Figure 6—figure supplement 1B*); (iii) additional replication origins may fire in the proximity of the stalled replication fork, which would explain the increase in both active phosphorylated helicase and DNA polymerases α and δ (*Figure 6—figure supplement 1C*). Having established the conditions in which DNA synthesis but not DNA unwinding is blocked and concomitantly polymerases are accumulated, we then addressed the consequences for chromatin motion (*Figure 6—figure supplement 2*).

## Accumulation of replisome components but not processive DNA synthesis per se restricts chromatin motion

To elucidate the roles of the processive DNA synthesis and loading of synthetic machinery in chromatin motion decrease in S-phase, we imaged single cells for PCNA, RPA34, and Cy3-dUTP pre- and post-aphidicolin treatment (*Figure 6A*, see also *Videos 4 and 5*). The PCNA and RPA patterns did not change in G1/G2, whereas in S-phase the PCNA was dissociated from chromatin and RPA was

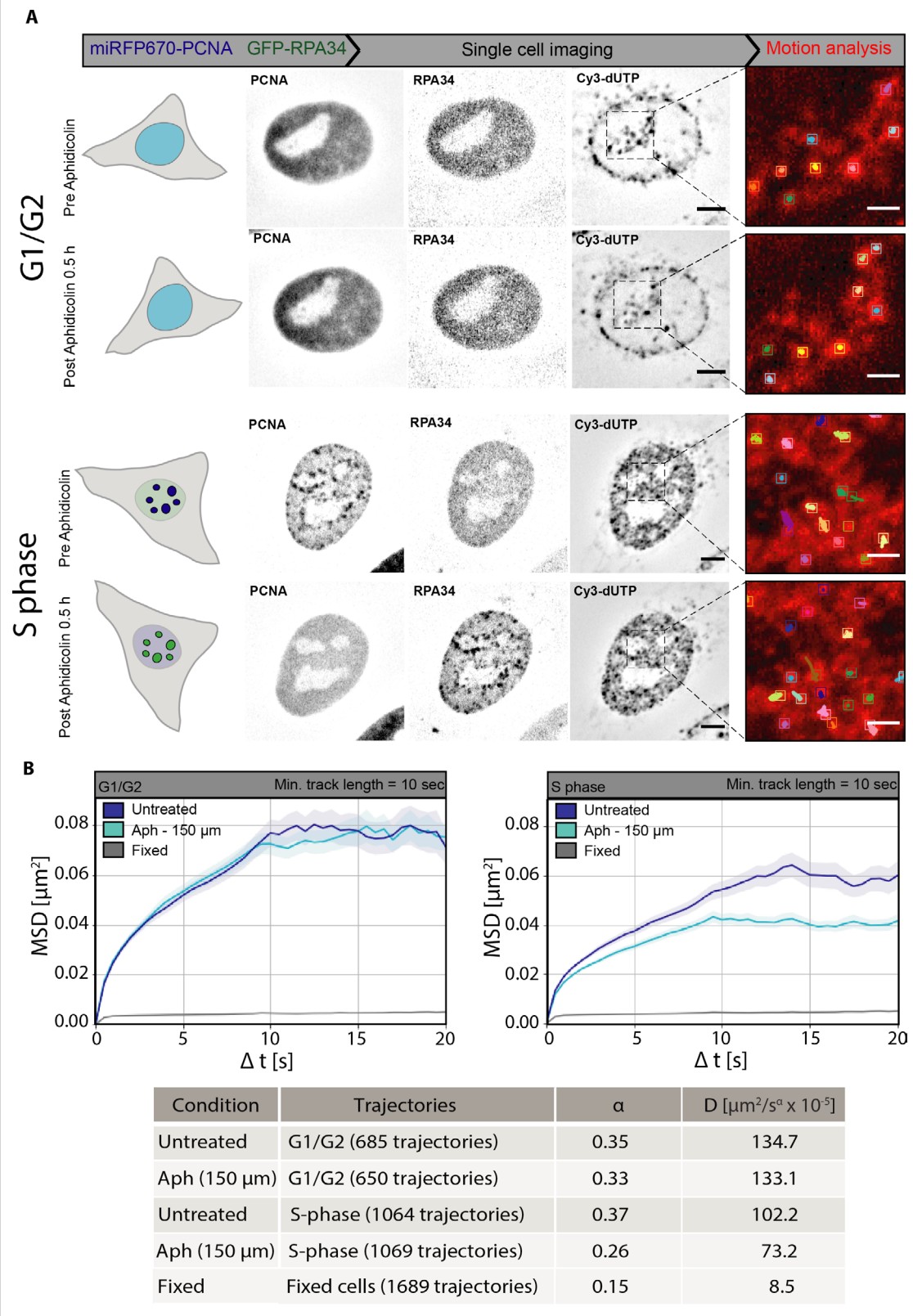

| Condition | Trajectories | α | D [μm²/sᵅ x 10⁻⁵] |
|---|---|---|---|
| Untreated | G1/G2 (685 trajectories) | 0.35 | 134.7 |
| Aph (150 μm) | G1/G2 (650 trajectories) | 0.33 | 133.1 |
| Untreated | S-phase (1064 trajectories) | 0.37 | 102.2 |
| Aph (150 μm) | S-phase (1069 trajectories) | 0.26 | 73.2 |
| Fixed | Fixed cells (1689 trajectories) | 0.15 | 8.5 |

**Figure 6.** Inhibition of DNA synthesis by aphidicolin further restricts chromatin mobility in S-phase but not in G1/G2 cells. (**A**) Representative images of HeLa GFP-RPA34 cells transfected with a construct coding for miRFP670-proliferating cell nuclear antigen (PCNA) and Cy3-dUTP nucleotides for both G1/G2 and S-phase cells pre- and post-aphidicolin (Aph) treatment. The chromatin foci were imaged using the spinning disk microscope. The image sequences were used to perform motion analysis. The cropped region (black dashed lines) shows the motion analysis of chromatin tracks before and

*Figure 6 continued on next page*

*Figure 6 continued*

after treatment of the same cells (see also *Videos 6 and 7*). (**B**) Mean square displacement (MSD) curves over time were plotted for all chromatin tracks for untreated and aphidicolin-treated (150 µM) cells in G1/G2 and S-phase. The error bars are represented in transparent color around the curve. The table below provides the detailed information on number of trajectories per individual sample along with average diffusion rates (µm²/s × 10⁻⁵) and anomalous α coefficient showing subdiffusion. The MSD were plotted with error bars (standard deviation) represented in transparent color around the curve. Scale bar: 5 µm. Insets scale bar: 1 µm.

The online version of this article includes the following figure supplement(s) for figure 6:

**Figure supplement 1.** Model describing the different scenarios of replisome response to stress.

**Figure supplement 2.** Graphical abstract showing chromatin dynamics during S-phase and replication stress.

accumulated at the same previously replicating sites (*Figure 6A*, see also *Videos 6 and 7*). Chromatin motion analysis was performed on DNA labeled with Cy3-dUTP for both G1/G2 cells and S-phase cells pre- and post-treatment with aphidicolin. We observed that chromatin motion was unaffected in G1/G2, which fit with our prediction, as in G1/G2 there is no active DNA synthesis besides possible DNA repair processes on a limited genomic scale (*Figure 6B*). As hydroxyurea, another DNA synthesis inhibitor, significantly affected chromatin mobility outside of S-phase, we did not further pursue it. Surprisingly, aphidicolin treatment and inhibition of DNA synthesis led to additional decrease in chromatin motion (*Figure 6B*) and the chromatin became even more constrained than at the proximity of the active replication sites in S-phase cells (see *Figure 4C*). As quantified above, after aphidicolin treatment, the helicases were still loaded and actively spooling DNA through, whereas the DNA polymerases α and δ albeit not synthesizing DNA accumulated on chromatin at the sites of helicase/RPA accumulation (*Figure 5*).

In summary, we propose that the accumulation of the helicase and polymerase complexes on chromatin together with the continuous loading of the ssDNA binding protein (RPA) covering the ssDNA strands stiffens the DNA polymer and restricts its diffusional motion. This study provides new insights on the kinetics of DNA replication proteins loading upon DNA replication stress and elucidates the transient and localized immobilization of chromatin during DNA replication.

# Materials and methods
## Cells
All cells used were tested and negative for mycoplasma and authenticated via STR profiling (ATCC). Mycoplasma test was performed using PCR technique for amplification of mycoplasm specific DNA. The following primers are used: **myco-fw** 5'-TGCACCATCTGTCACTCTGTTAACCTC **myco-rv** 5'-G GGAGCAAACAGGATTAGATACCCT and 2 ml culture supernatant was used as template. Human cervical cancer cell line HeLa Kyoto (*Erfle et al.,*

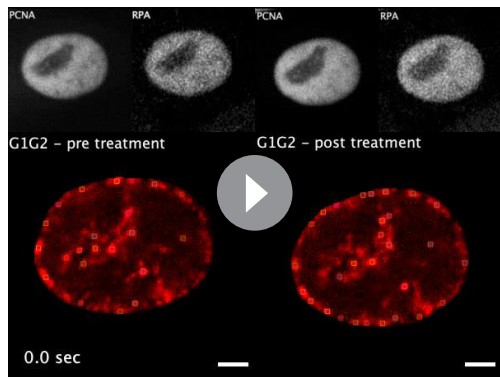

**Video 6.** Time-lapse microscopy of HeLa K cells pre- and post-aphidicolin treatment in G1/G2 phase expressing fluorescent proliferating cell nuclear antigen (PCNA) and RPA and labeled chromatin (red). Scale bar: 5 µm.

https://elifesciences.org/articles/87572/figures#video6

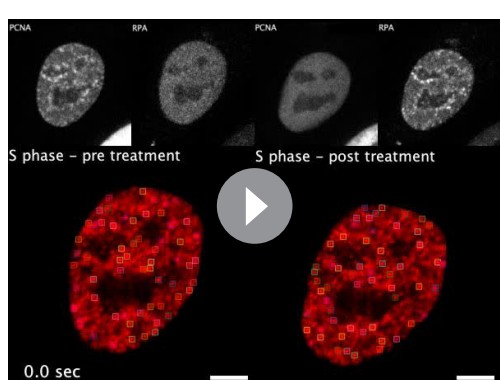

**Video 7.** Time-lapse microscopy of HeLa K cells pre- and post-aphidicolin treatment in S-phase expressing fluorescent proliferating cell nuclear antigen (PCNA) and RPA and labeled chromatin (red). Scale bar: 5 µm.

https://elifesciences.org/articles/87572/figures#video7

*2007*) provided by Jan Ellenberg (EMBL, Heidelberg, Germany) and human normal diploid fibroblasts from lung tissue IMR90 (*Nichols et al., 1977*) provided by Argyris Papantonis (Georg-August-Universitat Gottingen, Germany) were used in the study. Previously published HeLa Kyoto cells expressing GFP-PCNA (*Chagin et al., 2016*) fusion protein were used to monitor cell cycle progression. HeLa Kyoto GFP-RPA34 were generated using the Flp-In recombination system based on the Flp site-specific recombinase (Cat.No.: K6010-01, Invitrogen, Waltham, MA, USA). The HeLa Kyoto FRTLacZ cells containing a genomically integrated FRT site described earlier (*Chagin et al., 2016*) were cotransfected with pFRT-B-GRPA34 (*Supplementary file 1b*) (encoding GFP-RPA34) and pOG44 Flp-recombinase using Neon transfection (Cat.No.: MPK5000, Invitrogen, Waltham, MA, USA). Four hours after transfection the cell culture medium was exchanged and cells were grown for 48 hr and selected with 2.5 mg/ml blasticidin (Cat.No.: R210-01, Invitrogen, Waltham, MA, USA). A stable monoclonal line was isolated using blasticidin selection. All cells were maintained in Dulbecco's modified Eagle medium high glucose (Cat.No.: D6429, Sigma-Aldrich Chemie GmbH, Steinheim, Germany) supplemented with 10% fetal calf serum, 1× glutamine (Cat.No.: G7513, Sigma-Aldrich, St Louis, MO, USA) and 1 µM gentamicin (Cat.No.: G1397, Sigma-Aldrich, St Louis, MO, USA) in a humidified atmosphere with 5% $CO_2$ at 37°C. Additional experiments confirmed that the transgenic gene product co-localized with the endogenous protein (not shown) and was present at sites of active replication (*Figure 5—figure supplement 3*). The culture medium was changed every day and cells were split every 2 days. Cell line characteristics are summarized in *Supplementary file 1a*.

To block DNA replication, cells were treated with aphidicolin (Cat.No.: A0781-1MG, Sigma-Aldrich, St Louis, MO, USA) at final concentration of 150 µM (*Supplementary file 1c*). Cells were subsequently examined for 30 min (aphidicolin) following drug exposure. To confirm that DNA synthesis was inhibited, cells were labeled with 10 µM nucleoside analog 5-ethynyl-2'-deoxyuridine (Cat.No.: 7845.1, ClickIt-EdU cell proliferation assay, Carl Roth, Karlsruhe, Germany) (*Supplementary file 1c*) in media for 10 min to evaluate the extent of replication in control and treated cells (*Figure 5—figure supplement 1*).

For synchronization of HeLa cells, the cells were seeded on tissue culture dishes at high confluency. Once the cells were confluent, the cells were placed on a shaker for 5 min. The detached mitotic cells were collected from the supernatant and seeded on coverslips. Once the cells were in G1, they were fixed and stained with DAPI for quantification (*Figure 3—figure supplement 1B*).

## Live cell imaging and replication labeling

For live cell microscopy, cells were transfected using a Neon transfection system (Cat.No.: MPK5000, Invitrogen, Waltham, MA, USA). Briefly, the asynchronous population of cells were washed with 1× phosphate-buffered saline (PBS)/EDTA, trypsinized, and collected in a 15 ml tube. The cells were pelleted at 300 × *g* for 5 min. The media was removed and cells were resuspended in 100 µl resuspension buffer R and transferred to a 1.5 ml microcentrifuge tube. Either 15 µg of plasmid DNA or/ and 0.5 µl (25 nM) Cy3-dUTP (Cat.No.:ENZ-42501, Enzo Life Sciences, Lörrach, Germany) was added to the cell mixture (*Supplementary file 1b and c*). The Neon tip was immersed into the cell mixture and the mixture pipetted taking care to avoid bubbles. The tip was immersed in electrolytic buffer E2 and cells were electroporated (HeLa [voltage – 1005 V, width – 35, pulses – 2], IMR90 [1100 V, width – 30, pulses – 1]). The electroporated mixture was transferred to Ibidi µ-dish chambers (Cat.No.: 80826, Ibidi, Gräfelfing, Germany). Additionally, IMR90 cells were transfected with miRFP670-PCNA plasmid (*Supplementary file 1b*) to mark the DNA replication sites. After transfection, cells were allowed to attach overnight and were imaged the next day. All imaging was performed at 37°C with a humidified atmosphere of 5% $CO_2$ using an Olympus environmental chamber (spinning disk microscope, *Supplementary file 1e*).

## Immunofluorescence

For immunofluorescence, cells were fixed with 3.7% formaldehyde/1× PBS (Cat.No.: F8775, Sigma-Aldrich Chemie GmbH, Steinheim, Germany) for 15 min and permeabilized with 0.7% Triton-X100 in 1× PBS for 20 min. All washing steps were performed with PBS-T (1× PBS/0.075% Tween-20). For detection of PCNA, cells were further incubated for 5 min in ice-cold methanol for antigen retrieval. Blocking (1% bovine serum albumin in 1× PBS) was performed for 30 min at room temperature. EdU was detected using the Click-IT assay as described by the manufacturer (1:1000 6-FAM azide

or 1:2000 5/6-sulforhodamine azide; Cat.No.: 7806 and 7776, respectively, Carl Roth, Karlsruhe, Germany). Primary and secondary antibodies were diluted in the blocking buffer and incubated for 1 hr at room temperature with subsequent 3×10 min of PBS-T washing. DNA was counterstained with DAPI (4',6-diamidino-2-phenylindole, 10 µg/ml, Cat.No.: D27802, Sigma-Aldrich Chemie GmbH, Steinheim, Germany) for 10 min, and samples were mounted in Vectashield (Cat.No.: VEC-H-1000, Vector Laboratories, Inc, Burlingame, CA, USA). Antibody characteristics are summarized in *Supplementary file 1d*.

## Western blot and chromatin fractionation

Cells for western blot were washed with 5 ml ice-cold 1× PBS once and 2 ml of ice-cold 1× PBS was added and cells were scraped using a cell scraper. Cells were then centrifuged in a 15 ml tube at 500 × *g* for 5 min. Cells were lysed for total cell lysates for 1 hr at 4℃ using the IP lysis buffer with 150 mM NaCl (Cat.No.: 0601.2, Carl Roth, Karlsruhe, Germany), 200 mM TrisCl pH 8 (Cat.No.: A1086.500, Diagonal, Münster, Germany), 5 mM EDTA (Cat.No.: 8040.2, Carl Roth, Karlsruhe, Germany), 0.5% NP-40 (Cat.No.: 74385, Sigma-Aldrich Chemie GmbH, Steinheim, Germany) and protease and phosphatase inhibitors PMSF (Cat.No.: 6367.1, Carl Roth, Karlsruhe, Germany), PepA (Cat.No.: 2936.2, Carl Roth, Karlsruhe, Germany), NaF (Cat.No.: 67414-1-ML-F, Sigma-Aldrich Chemie GmbH, Steinheim, Germany), $Na_3VO_4$ (Cat.No.: S6508-10G, Sigma-Aldrich Chemie GmbH, Steinheim, Germany). Protein fractionation of control and treated samples was performed as described in *Gillotin, 2018*. Briefly, equal number of cells were washed with buffer E1 (cytoplasmic fraction) and centrifuged at 1200 × *g* for 2 min and collected into a new tube. The step was repeated two times to remove excess cytoplasmic fraction. The pellet was then washed with buffer E2 (nucleoplasm fraction) and collected into a new tube. The chromatin fraction was isolated with buffer E3 and 1:1000 benzonase for 20 min at 25℃.

All lysates were then centrifuged at 13,000 rpm for 20 min at 4℃. The supernatant was collected into a new 1.5 ml tube and protein concentration was measured using the bovine serum albumin protein standard assay (Cat.No.: 23208, Thermo Fisher Scientific, Waltham, MA, USA) according to the manufacturer's protocol. 10% SDS-PAGE gel was prepared and 50 µg of protein lysate was loaded along with the protein standard ladder (Cat.No.: P7719S, New England Biolabs, Ipswich, MA, United States), and electrophoresis was performed for 1.5 hr in ice-cold 1× Laemmli electrophoresis running buffer. Then, the protein was transferred to the 0.2 µm nitrocellulose membrane using a semi-dry transfer system (#1703940, Trans-Blot SD Semi-Dry Transfer Cell, Bio-Rad, Hercules, CA, USA) for 55 min at 25 V using 1× transfer buffer (Pierce Western Blot Transfer Buffer 10×, Thermo Fisher Scientific, Waltham, MA, USA). After the transfer, the blotting membrane was incubated in a blocking buffer (5% low-fat milk in 1× PBS) for 30 min. The primary antibodies (*Supplementary file 1d*) were diluted in blocking buffer to 5% milk and incubated at 4℃ overnight. The next day the membrane was washed three times with 1× PBS-T (0.075 %) 10 min each. The membrane was then incubated with secondary antibodies (*Supplementary file 1d*) for 1 hr at room temperature. The membrane was washed again with 1× PBS-T (0.075 %) three times 10 min each and incubated with 1:1 ECL chemiluminescence solution (Clarity Western ECL, #170-5061, Bio-Rad Laboratories, Hercules, CA, USA). Signal was detected using an Amersham AI600 imager (*Supplementary file 1e*).

## Microscopy

Live cell imaging for chromatin mobility measurements were performed using the PerkinElmer UltraVIEW VoX system with a 60×/1.45 numerical aperture plan-apochromatic oil immersion objective. Cy3 and GFP were excited sequentially using 543 nm and 488 nm solid-state diode laser lines to minimize crosstalk. The standard protocol for examining chromatin mobility in Cy3-dUTP-labeled nuclei was as follows: first, a reference image comprising the miRFP670/GFP-PCNA, Cy3-dUTP, and the phase-contrast signal was collected from a single focal plane corresponding to the middle of the nucleus. This image demarcated the nuclear boundary, provided cell cycle information, and, in the case of S-phase cells, allowed us to correlate the positions of Cy3-dUTP foci with sites of DNA replication. Second, while maintaining the same focal plane, a time series (30–60 s) at a frame rate of 500 ms was captured. To maximize the temporal resolution, the time series consisted solely of the Cy3-dUTP channel and a PCNA reference frame at the beginning to obtain information on the cell cycle stage.

Multiple point time-lapse microscopy was performed using the multi-time option available in the spinning disk Volocity 6.3 software to image the chromatin (Cy3-dUTP) of the same cells pre- and post-treatment of aphidicolin. To minimize photo-toxicity over the course of the experiment, transmitted light contrast imaging was used to focus the cells. Live cell imaging was performed by following cells through the cell cycle and G1 and G2 stages were classified based on the previous cell cycle stage.

For the inhibition experiments (aphidicolin) different cells/points were chosen using the multipoint function of the Perkin Elmer spinning disk, and image sequences before the treatment were acquired. The reference image consisted of GFP-RPA34, Cy3-dUTP, and miRFP670-PCNA using 488 nm, 561 nm, and 640 nm solid-state diode lasers, respectively. After acquiring the reference images, the media containing the small molecule inhibitor was added to cells on the microscope for the required time and after treatment image sequences were acquired for analysis of chromatin motion.

High-throughput imaging was performed using the 40×/0.95 numerical aperture air objective of the PerkinElmer Operetta system. We used different filters (excitation/emission: 360/400, 460/490, 560/580) to image DAPI, EdU, and different replication proteins (*Figure 5*, *Figure 5—figure supplement 2*, *Supplementary file 1e*).

## Quantification of DNA synthesis inhibition

The high-throughput images were used to quantify the percentage of cells with inhibition of DNA synthesis upon aphidicolin treatment. A minimum of 100 fields with around 2000–5000 cells were acquired in all channels. The images were then analyzed using the PerkinElmer Harmony software. The steps in brief (*Figure 5—figure supplement 1*, *Figure 5—figure supplement 2*) include segmentation of nuclei using cell types of specific parameters like the diameter, splitting coefficient, and intensity threshold. The segmentation was then validated by visually checking it in randomly selected regions. Once the nuclei were segmented, cells touching the border were omitted. The intensity values with mean, median, standard deviation, and the sum of the intensities were obtained for individual cells. The datasheets were then imported to R and plots were generated. EdU signal was used to identify the population of cells actively replicating upon aphidicolin treatment (*Figure 5A*). The background intensity for EdU staining was determined using a negative control which was not treated with EdU but stained. The cells showing a mean intensity greater than the background intensity were separated into an EdU positive population and plotted.

## DNA quantification of labeled chromatin

DNA quantification of the labeled foci was done by automated image analysis. Image sequences with labeled chromatin were acquired on a Ultra-View VoX spinning disk microscope, using a 60× objective (*Figure 2*, *Figure 2—figure supplement 1*). For segmentation of replication foci, we used the protocol originally described in *Chagin et al., 2016*; *Chagin et al., 2015*. The channels comprising DAPI replication foci signals were imported into the software Perkin Elmer Volocity 6.3 and converted into volumes. The pixel dimensions of the images were set to the specifications for the spinning disk (x/y: 0.066 µm and z: 0.3 µm). The following processing steps were applied: Find objects ('nucleus') using the DAPI channel, method 'Intensity' (set manually to the optimal value), use fill holes in object/dilate/erode until the object optimally fits the nucleus, exclude objects by size <500 µm³. Find objects using the label channel, method 'Intensity' (lower limit: 1, upper limit: 65535), separate touching objects, exclude 'foci'' not touching 'nucleus'. Using the detected foci, the DNA content of foci was determined via the sum of intensities in the DAPI channel and the genome size of the cell type (*Figure 2*, *Figure 2—figure supplement 1*).

## Automated tracking of chromatin structures in time-lapse videos

The motility of fluorescently labeled chromatin structures in live cell fluorescence microscopy images was quantified within manually segmented single nuclei. The background image intensity was adjusted for each image sequence to the computed mean intensity value over all time points within a manually selected region of interest (ROI) of the background. Automatic tracking of multiple fluorescently labeled chromatin structures was performed using a probabilistic particle tracking approach, which is based on Bayesian filtering and multi-sensor data fusion (*Ritter et al., 2021*). This approach combines Kalman filtering with particle filtering and integrates multiple measurements by separate sensor models and sequential multi-sensor data fusion. Detection-based and prediction-based measurements

are obtained by elliptical sampling (*Godinez and Rohr, 2015*), and the separate sensor models allow considering different uncertainties. In addition, motion information based on displacements from past time points is exploited and integrated in the cost function for correspondence finding. Chromatin structures are detected by the spot-enhancing filter (SEF) (*Sage et al., 2005*) which consists of a Laplacian-of-Gaussian filter followed by thresholding the filtered image and determination of local maxima. The threshold is automatically determined by the mean of the absolute values of the filtered image plus a factor times the standard deviation. We used the same threshold factor for all images of an image sequence (*Figure 3—figure supplement 1*).

## Chromatin motility analysis

Based on the computed trajectories, the motility of chromatin structures was analyzed, and the motion type was determined for different cell cycle stages along with active replication sites, and inhibition of DNA synthesis with aphidicolin. We performed an MSD analysis (*Saxton, 1997*) and computed the MSD as a function of the time interval Δt for each trajectory (*Figure 3—figure supplement 1*). The MSD curves for all trajectories with a minimum time duration of 10 s (corresponding to 20 time steps) under one condition were averaged. We considered only the trajectories with a time duration larger than the minimum time duration which improved the accuracy of the motility analysis. We fitted the anomalous diffusion model to the calculated MSD values to obtain the anomalous diffusion coefficient α. The motion was classified into confined diffusion, obstructed diffusion, and normal diffusion (*Bacher et al., 2004*). To determine the diffusion coefficient D (μm²/s), the diffusion model was fitted to the MSD values. In case of IMR90 cells, affine image registration was performed using the method in *Celikay et al., 2022*, to address the stronger cell movement compared to HeLa cells.

## CCD/proximity analysis

Automatic proximity analysis of chromatin and PCNA was performed using the computed trajectories of chromatin structures and detected sites of active DNA synthesis represented by fluorescently labeled PCNA. Only trajectories of chromatin structures present at the first time point of an image sequence and with a minimum time duration of 10 s (corresponding to 20 time steps) were considered. PCNA foci were automatically detected in the fluorescence microscopy images by the SEF (*Figure 3—figure supplement 1*). For each PCNA image, a single nucleus was manually segmented, and the background intensity was adjusted to the computed mean intensity value within a manually selected ROI of the background. Proximity was determined for the first time point of the trajectory of a chromatin structure and detected PCNA foci using a graph-based k-d-tree approach (*Bentley, 1975*). Due to the k-d-tree structure, this approach allows efficient computation of the nearest neighbor query based on the Euclidean distance between foci in the chromatin and PCNA channel. If a chromatin structure at the first time point of the image sequence has a nearest PCNA neighbor within a maximum distance, the trajectory of a chromatin structure is considered within CCD. Otherwise, the trajectory is considered outside the CCD (*Figure 4—figure supplement 1*).

## Accumulation analysis

To analyze the focal RPA accumulation upon DMSO/aphidicolin treatment, cell nuclei were segmented using the Volocity software (Version 6.3, Perkin Elmer). The GFP-RPA34 signal was segmented before and after treatment of the same cell in the live experiments and plotted over time after DMSO and drug treatment. The GFP-RPA intensities were measured and the coefficient of variation cV = σ/μ, with σ = standard deviation and μ=mean, was calculated for all time points (*Figure 5—figure supplement 4*, *Figure 5—figure supplement 5*). All values were normalized to the DMSO treatment cV = cV(tpx)/cV(tp0) with tpx: any given time point imaged, tp0: pretreatment time point and plotted using RStudio (*Supplementary file 1i*).

## High-throughput image analysis of replisome components

The images from the Nikon crest Ti2 system were analyzed with the custom-made image analysis pipeline in KNIME Analytics Platform. The image analysis pipeline was constructed as follows (*Figure 5—figure supplement 7*, *Figure 5—figure supplement 8*). Briefly, the channels were separated. The DAPI channel was used for the nuclei segmentation. Nuclei were segmented based on manually chosen intensity threshold, the Watershed Transform was applied next to separate the

close-positioned nuclei. The segmented nuclei were converted into a mask with each nucleus DAPI intensity and texture features recorded. The nuclei population was further thresholded by nucleus area and circularity to eliminate segmentation artifacts. The EdU and replication protein channels were subjected to foci segmentation based on a wavelet transform algorithm. The algorithm parameters were selected individually for each type of the replication protein and maintained the same between the control and treated samples. The nuclear mask and EdU foci/replication protein foci masks were overlaid to filter only the foci inside the nuclear areas. The EdU foci/replication protein foci intensity parameters (total focus intensity, mean focus intensity), area, and foci number per nucleus were exported as XLSX files for further analysis. The data was analyzed in RStudio (https://posit.co/download/rstudio-desktop/). First, the S-phase cell population was identified by the number of EdU foci per nucleus. The EdU foci number threshold was set as 50 for the cells in control samples, and 55 for aphidicolin-treated samples among all datasets. The nuclei in S-phase were next analyzed for their replication protein accumulation. The total levels of the replication proteins were plotted as box plots, ***$p<0.001$ by Wilcoxon rank-sum test, for aphidicolin-treated vs. control sample.

## Acknowledgements

We thank Anne Lehmkuhl and Diana Imblan for their excellent technical support. We are thankful to Alexander Rapp, Hector Romero, and Gaudenz Danuser for their valuable suggestions. We also thank Argyris Papantonis (Georg-August-Universität Göttingen, Germany) for providing the IMR90 cells. Funding. This research was funded by the Deutsche Forschungsgemeinschaft (DFG, German Research Foundation) – Project-ID 393547839 – SFB 1361, CA 198/9-2 Project-ID 232488461 and CA 198/15-1 (SPP 2202) Project-ID 422831194 to MCC; and RO 2471/10-1 (SPP 2202) Project-ID 402733153 and SFB 1129 (project Z4) Project-ID 240245660 to KR.

## Additional information

### Funding

| Funder | Grant reference number | Author |
|---|---|---|
| Deutsche Forschungsgemeinschaft | Project-ID 393547839 - SFB 1361 | M Cristina Cardoso |
| Deutsche Forschungsgemeinschaft | CA 198/9-2 Project-ID 232488461 | M Cristina Cardoso |
| Deutsche Forschungsgemeinschaft | CA 198/15-1 (SPP 2202) Project-ID 422831194 | M Cristina Cardoso |
| Deutsche Forschungsgemeinschaft | RO 2471/10-1 (SPP 2202) Project-ID 402733153 | Karl Rohr |
| Deutsche Forschungsgemeinschaft | SFB 1129 (project Z4) Project-ID 240245660 | Karl Rohr |

The funders had no role in study design, data collection and interpretation, or the decision to submit the work for publication.

### Author contributions

Maruthi Kumar Pabba, Conceptualization, Data curation, Formal analysis, Investigation, Visualization, Writing - original draft; Christian Ritter, Data curation, Software, Formal analysis, Writing – review and editing; Vadim O Chagin, Conceptualization, Supervision, Investigation, Writing – review and editing; Janis Meyer, Kerem Celikay, Alice Kristin Schmid, Software; Jeffrey H Stear, Investigation, Writing – review and editing; Dinah Loerke, Software, Writing – review and editing; Ksenia Kolobynina, Formal analysis; Paulina Prorok, Investigation; Heinrich Leonhardt, Conceptualization, Writing – review and editing; Karl Rohr, Conceptualization, Supervision, Funding acquisition, Project administration, Writing – review and editing; M Cristina Cardoso, Conceptualization, Resources, Supervision, Funding acquisition, Project administration, Writing – review and editing

## Author ORCIDs
Maruthi Kumar Pabba ⓘ http://orcid.org/0000-0003-2855-1392
M Cristina Cardoso ⓘ https://orcid.org/0000-0001-8427-8859

Joint Public Review: https://doi.org/10.7554/eLife.87572.3.sa1
Author Response https://doi.org/10.7554/eLife.87572.3.sa2

## Additional files

### Supplementary files
• Supplementary file 1. The *Supplementary file 1* contains tables 1a–1i, which describes the materials, software, and datasets.

### Data availability
All data are available from the OMERO open microscopy environment public repository and TUdatalib (https://doi.org/10.48328/tudatalib-873). All renewable biological materials will be made available upon request from the corresponding author M. Cristina Cardoso (cardoso@bio.tu-darmstadt.de).

The following dataset was generated:

| Author(s) | Year | Dataset title | Dataset URL | Database and Identifier |
|---|---|---|---|---|
| Pabba MK, Ritter C, Chagin V, Stear J, Loerke D, Prorok P, Schmid AK, Leonhardt H, Rohr K, Cardoso C | 2022 | Replisome loading reduces chromatin motion independent of DNA synthesis | https://doi.org/10.48328/tudatalib-873 | TUdatalib, 10.48328/tudatalib-873 |

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
