## [Editor Report · eLife assessment]

This is a **valuable** investigation of the chromatin dynamics throughout the cell cycle by using fluorescence signals and patterns of GFP-PCNA and CY3-dUTP, which labels newly synthesized DNA. The authors report reduced chromatin mobility in S relative to G1 phase. The technology and methods used are **solid**. The data will be of interest to researchers working on chromatin dynamics.

---

## [Referee Report · Joint Public Review]

The manuscript presented by Pabba et al. studied chromatin dynamics throughout the cell cycle. The authors used fluorescence signals and patterns of GFP-PCNA (GFP tagged proliferating cell nuclear antigen) and CY3-dUTP (which labels newly synthesized DNA but not the DNA template) to determine cell cycle stages in asynchronized HeLa (Kyoto) cells and track movements of chromatin domains. PCNA binds to replication forks and form replication foci during the S phase. The major conclusions are: (1) Labeled chromatin domains were more mobile in G1/G2 relative to the S-phase. (2) Restricted chromatin motion occurred at sites in proximity to DNA replication sites. (3) Chromatin motion was restricted by the loading of replisomes, independent of DNA synthesis. This work is based on previous work published in 2015, entitled "4D Visualization of replication foci in mammalian cells corresponding to individual replicons," in which the labeling method was demonstrated to be sound.

Comments on latest version: The revised manuscript has included data from a diploid cell line IMR90 (fibroblasts isolated from normal lung tissue) to strengthen the conclusions. Overall, quality of the work is substantially improved.

---

## [Author Response]

The following is the authors’ response to the original reviews.

**eLife assessment**
This is a valuable investigation of the chromatin dynamics throughout the cell cycle by using fluorescence signals and patterns of GFP-PCNA and CY3-dUTP, which labels newly synthesized DNA. The authors report reduced chromatin mobility in S relative to G1 phase. The technology and methods used are solid, but the significance of the work is reduced by the model system employed, the HeLa cell line, which has a greatly abnormal genome.

We have obtained data from a diploid human cell that validates the reduction of S-phase chromatin mobility.

**Public Review:**
The manuscript presented by Pabba et al. studied chromatin dynamics throughout the cell cycle. The authors used fluorescence signals and patterns of GFP-PCNA (GFP tagged proliferating cell nuclear antigen) and CY3-dUTP (which labels newly synthesized DNA but not the DNA template) to determine cell cycle stages in asynchronized HeLa (Kyoto) cells and track movements of chromatin domains. PCNA binds to replication forks and form replication foci during the S phase. The major conclusions are: (1) Labeled chromatin domains were more mobile in G1/G2 relative to the S-phase. (2) Restricted chromatin motion occurred at sites in proximity to DNA replication sites. (3) Chromatin motion was restricted by the loading of replisomes, independent of DNA synthesis. This work is based on previous work published in 2015, entitled "4D Visualization of replication foci in mammalian cells corresponding to individual replicons," in which the labeling method was demonstrated to be sound. Although interesting, reduced chromatin mobility in S relative to G1 phase is not new to the field.

It was first shown in yeast (Heun et al. 2001; DOI:10.1126/science.1065366) that the S-phase mobility is reduced compared to the G1 phase. This was followed by other papers showing the same in yeast [(Gasser 2002; DOI:10.1126/science.1067703), (Smith et al. 2019; DOI: 10.1091/mbc.E19-08-0469)]. The relation between chromatin motion and cell cycle progression in the mammalian genome is less studied. Over recent years there have been a few studies that addressed chromatin mobility and cell cycle progression but from a different perspective. In the publication Nozaki et al. (2017;DOI:10.1016/j.molcel.2017.06.018) chromatin motion analysis was performed on single histones. The study did not find a significant change of histone/nucleosome mobility measured during cell cycle progression. Using CRISPR/dCas9 to label random DNA loci, Ma et al. (2019; DOI:10.1083/jcb.201807162) found that chromatin motion in S-phase was significantly lower than in the G1 phase. However, most of the studies measure the chromatin motion using either insertion of ectopic loci or proteins marking the loci (dCas9) or histones. Using either ectopic loci addition or CRISPR/dCas9 might have an effect on the chromatin mobility itself and measuring single histone motion is not equivalent to measuring the motion of DNA segments. We, therefore, opted to label the DNA directly using the replication of the DNA. In this manner we preserve the native chromatin structure and, thus, motion.

Importantly, in addition to measuring decreased DNA motion in S-phase, our study indicates that it is not the DNA synthesis per se but the loading of replisomes onto chromatin that slows down its motion. This allowed us to propose a mechanism on how chromatin motion is affected by DNA replication in S-phase.

The genome in HeLa cells is greatly abnormal with heterogeneous aneuploidy, which makes quantification complicated and weakens the conclusions.

We agree that the HeLa cells are aneuploid and we have addressed the heterogeneity of HeLa Kyoto within our detection methods (for clarification see point 3). To validate our conclusions in normal diploid human cells, we performed the chromatin mobility analysis using human fibroblasts (IMR90 cells in figures 2, 3 and S2) and plotted the MSD curves for different cell cycle stages. The outcome of this analysis showed that the mobility of chromatin in diploid fibroblasts in S-phase is lower than in G1 and G2. In fact, this effect is stronger in IMR90 cells than in HeLa Kyoto cells. Hence, this is not an aneuploid tumor cell phenomenon.

The manuscript is difficult to follow in places due to insufficient clarity. The manuscript should be written in a way that can be understood without referencing previous articles. Overall, the work is moderately impactful to the field.Major recommendations:1. In Figure 1B, the illustration and images for S phase are confusing. The author should specify which is early S and which is late S. Do the yellow circles represent GFP-PCNA foci? How did the authors distinguish mid S from early S and late S (in Figure 2)? Are all images in Figure 1 scaled to the same contrast threshold?

The yellow circles correspond to the colocalized signal of GFP-PCNA and Cy3-dUTP that overlap and represent the labeled chromatin sites that are replicated in the next cell cycle.

We clarified all the points mentioned above and updated figure 1 and figure 2 accordingly.

2. In Figure 2B, the y-axis is marked as "Frequency of cells" but the equation listed below is counting DNA (per focus). How to convert DNA (per focus) to DNA (per cell)? The x-axis is marked as "Genome size" without any unit (e.g., kb? Mb?) The x-axis seems to be the C factor, not the genome size.

To determine the amount of DNA present in each labeled DNA focus, we first segmented the whole nucleus and measured the total intensity of DAPI (DNA amount) which is called IDNA TOTAL. Then the labeled replication foci are segmented and the intensity of label present in each segmented foci is measured (IRFi). Throughout the S-phase progression the amount of DNA increases twofold from early to late S-phase. The cells at each cell cycle stage were determined using the PCNA pattern. By plotting the frequency (number of cells) and the relative genome content normalized to the G1 stage we calculated the relative genome size otherwise called cell cycle correction factor for each stage from G1 to G2. The ratio of DNA intensity in labeled replication (IRFi)/ to the total DNA intensity of DAPI (IDNA total) gives the fraction of DNA present in each foci compared to the whole nucleus. This ratio was then multiplied by the genome size (Kbp) of HeLa Kyoto cells which was measured and published in Chagin et al. (2016; DOI:10.1038/ncomms11231). This gives us the approximate amount of DNA present in each labeled replication foci in Kbp. Since the genome duplicates over cell cycle stages, the measured DNA content in IRFi was corrected to the cell cycle stage (determined by PCNA) by multiplying the cell cycle correction factor.

3. HeLa cells are known to be highly heterogeneous and heavily aneuploidy. Cells in one sample have different numbers of chromosomes ranging from 50 - 80. Therefore, GS (genome size) for each cell should not be the same. Using one constant GS in the equation for every cell introduces errors. Has the cell-to-cell variation been considered and corrected in the data? If not, the authors should provide information regarding cell-to-cell variations, such as the intensity variation of nuclear DAPI signals in synchronized cells.

It is true that the HeLa genome is aneuploid. However, the heterogeneity of the genome is true, if one compares different HeLa strains as studied in Frattini et al. (2015; DOI:10.1038/srep15377), where they show the variability of genome and RNA expression profiles and small genomic rearrangements among different HeLa strains. However, to our knowledge, it is not studied extensively or shown whether the heterogeneity and aneuploidy would also be a cell to cell variation. Therefore, we performed a control experiment to verify the variability between HeLa Kyoto cells, where we either synchronized or not and stained with DAPI and the DNA content profiles of all cells were plotted as a histogram (supplementary figure 1B) to show that cell to cell variations is not present and by synchronizing, we see that the cell population in G1, has similar DNA content showing that the cell to cell variability is negligible in our detection methods. Nonetheless, we have obtained data using normal diploid human fibroblasts, which validated our outcome.

STABLE:

Macville, Merryn, et al. "Comprehensive and definitive molecular cytogenetic characterization of HeLa cells by spectral karyotyping." Cancer research 59.1 (1999): 141-150.

UNSTABLE:

Liu, Yansheng, et al. "Multi-omic measurements of heterogeneity in HeLa cells across laboratories." Nature biotechnology 37.3 (2019): 314-322.

Landry, Jonathan JM, et al. "The genomic and transcriptomic landscape of a HeLa cell line." G3: Genes, Genomes, Genetics 3.8 (2013): 1213-1224.

4. The chromatin foci are in a variety of sizes and intensities. How were boundaries of foci determined? Weak foci were picked up in one image but not in another. This is a concern because the size of the chromatin domain could influence mobility measurement. The authors should provide control experiments or better explanations for detecting and selecting chromatin foci.

The method for detecting chromatin foci is described in “Materials and Methods” section “Automated tracking of chromatin structures in time-lapse videos”. “Chromatin structures are detected by the spot-enhancing filter (SEF) (Sage et al., 2005; doi:10.1109/TIP.2005.852787) which consists of a Laplacian-of-Gaussian (LoG) filter followed by thresholding the filtered image and determination of local maxima. The threshold is automatically determined by the mean of the absolute values of the filtered image plus a factor times the standard deviation.” For reasons of consistency, we used the same threshold factor for all images of an image sequence. Therefore, depending on the intensity distribution in an image, it can happen that weak foci are not detected in some images. Alternatively, one could manually adapt the threshold factor for all single images, which, however, would be subjective. We now added the information that we used the same threshold factor for all images of an image sequence.

5. In Figure 3, the authors combined MSD from G1 and G2 in one group. Has any published data suggested that chromatin dynamics are the same in G1 and G2?

To clarify this we separated G1 and G2 mobility measurements in supplementary figure S2 and updated the figures and text accordingly.

6. In Figure 3B, cytoplasmic CY3-dUTP foci are found in the G1/G2 and S images. Are these CY3-dUTP aggregates? If so, are they also found in the nucleus? What is the mobility of the cytoplasmic CY3-dUTP foci?

These are aggregates and not found in the nucleus. These foci were excluded from the analysis by using a nuclear mask based on the PCNA signal. This information was added to the figure 3B legend.

7. In Figure 4, how is colocalization defined? 1.8 um is approximately the size of a chromosome territory, which is much larger than 0.5 Mb. Two foci that are 1.8 um apart should not be considered in the same chromosome.

We agree that colocalized would indeed mean that the signals are overlapping. Therefore, we updated the figures and text as center to center distance or proximity analysis.

Minor comments:1. Figure 3D should be presented by a box and whisker plot. The histogram does not show an actual distribution of the data.

The histograms shown in figure 3D is the average mean square displacement measurement value for different cell cycle stages. These are the same data shown in the table. Therefore, the histogram is removed and the table in figure 3C is retained.

2. Please explain Figure 3C error bars in the figure legend. Are they SD?

The error bars of the MSD curves (highlighted in bright color around the curves) in figure 3C show the standard error of the mean (SEM) representing the deviations between the MSD curves for an image sequence. We clarified this in the legend of Figure 3C.

3. In Figure 5C, some western blotting results seem to be assembled from replicate experiments. Comparing signals from one experiment with the same background is suggested.

We made sure that the western blots from the same replicates are cropped and the information is also added to the respective figure legends.